# Gut microbial signatures expose the westernized lifestyle of urban Ethiopian children
Lydia Kirsche [1] ✉, Peter Leary[1,2], Martin J. Blaser [3], Michael Scharl[4], Adugna Negussie [5] & Anne Müller [1] ✉

Gut microbiota composition has been extensively studied in European and North American pediatric cohorts, as well as in rural African children. Much less attention has been paid to urban African children, whose families have transitioned to a "Western" lifestyle characterized by smaller family sizes, access to perinatal care including C-section delivery, non-traditional food sources and widespread availability of antibiotics. We analyzed fecal samples from ~200 Ethiopian children aged 2-5 years from Adama, Ethiopia, using 16S rRNA gene sequencing and shotgun metagenomics. We found that well-studied factors such as delivery mode, breastfeeding and family size have only minor effects on α-diversity, whereas household crowding (single vs. multiple rooms) and consumption of the traditional fermented cereal *Eragrostis tef* predict higher α-diversity. Stunted growth and absence of Helicobacter pylori infection were additional factors associated with increased fecal microbial diversity. Metagenomic profiling revealed that rural African signature genera such as *Segatella* and *Prevotella* were largely absent; instead, urban Ethiopian children displayed a high Firmicutes/Bacteroidota ratio and enrichment of metabolic pathways linked to a westernized diet, resembling European rather than rural Ethiopian children. These results indicate that an urban westernized lifestyle alters gut microbiota composition, which may be partially offset by a traditional fermented diet.

The assembly of the human microbiota of the gut and other body sites starts at birth, usually by vertical transmission of microbes from mother to child[1–4]. The acquisition of microbes in early life is modulated by delivery mode, intrapartum antibiotic prophylaxis and breast vs. formula feeding[5–8]. Delivery by cesarean section in particular leads to a disruption of the transmission of maternal microbial strains and facilitates colonization by opportunistic pathogens associated with the hospital environment[7], which has been attributed to the delayed acquisition of Bacteroides species[6,7,9]. Mother-to-child transmission is more rapid and effective in home as opposed to hospital births, and delayed by cesarean section[10]. The presence and dominance of Bifidobacteria (*B. longum* and *B. breve*) in the pioneering neonatal gut microbiota is particularly effective in directing a stable community assembly trajectory and ensuring long-term pathogen colonization resistance, a process that is compromised by cesarean section[11]. After the post-natal period, perturbations and lifestyle conditions affecting the composition of the gut

microbiota in older infants and young children include exposure to antibiotics[9,12], and a Western diet and lifestyle[13]. Diet, age, geography, and population density are the main parameters affecting gut microbiota composition into, and in adulthood[14–16]. Migration represents another major factor; for instance, immigration from a non-Western country to the United States was associated with immediate loss of gut microbiome diversity and function, and the displacement of native strains[17].

Comparisons of non-Western and Western pediatric and adult cohorts have revealed major differences in microbiota composition that could be attributed to diet and lifestyle; for example, an early 16S rRNA-based comparison of children from rural Burkina Faso and urban Europe showed a significant enrichment of Bacteroidetes and depletion of Firmicutes in the former with a unique abundance of *Prevotella* and *Xylanibacter*[18]; an overrepresentation of *Prevotella*, especially *Prevotella copri* clades (and their underrepresentation in Western cohorts) was confirmed in rural adult cohorts from Ethiopia, Ghana, and Tanzania, and even detected in ancient

[1]Institute of Molecular Cancer Research, University of Zürich, Zürich, Switzerland. [2]Functional Genomics Center Zürich, University of Zürich and ETH Zürich, Zürich, Switzerland. [3]Center for Advanced Biotechnology and Medicine, Rutgers University, Piscataway, NJ, USA. [4]Department of Gastroenterology and Hepatology, University Hospital Zurich, University of Zurich, Zurich, Switzerland. [5]Department of Medical Laboratory Sciences, College of Health Sciences, Arsi University, Asella, Ethiopia. ✉e-mail: kirsche@imcr.uzh.ch; mueller@imcr.uzh.ch

human metagenomes[19]. Prevotellaceae were abundant and varied with the seasons, in Tanzanian hunter-gatherers adhering to a traditional lifestyle[20]. Time-resolved longitudinal analyses of the microbiome differences between rural Ethiopian and European infants suggest that differences increase with the age of the individuals, with newborns showing substantially less divergence than children and children showing less divergence than adults[13]. Also in this study, *Prevotella species* emerged as highly overrepresented in Ethiopian (as well as other African) rural pediatric cohorts and absent in European cohorts, and fermented foods were identified as a source of some of the microbial colonizers[13].

While substantial metagenomic information is thus available for European children on the one hand, and rural African children on the other (the above-mentioned studies from Burkina Faso, Ethiopia, Ghana, Tanzania, and also Zimbabwe)[18,19,21], much less attention has been paid to urban African children. Here, we fill this gap by reporting 16S rRNA gene and species-resolved shotgun metagenomic analyses conducted on >200 and >100 young children aged 2–5, respectively, that were born between 2018 and 2022 in Ethiopia's second-largest city, Adama. Microbial composition and diversity of our urban children was correlated with perinatal parameters, such as breast vs. formula feeding, but also lifestyle, health and clinical parameters, such as growth and weight gain trajectories, family size, crowding, the consumption of fermented foods and the *Helicobacter pylori* (*H. pylori*) status. Finally, the urban children's metagenomes were compared to those of their rural Ethiopian, and to European counterparts. We find that urban Ethiopian children harbor a microbiota that is more closely related to that of Western European children than to that of rural Ethiopians, with the notable absence of *Prevotella*, *Segatella* and other species associated with a non-Western, traditional lifestyle.

## Results

### 16S rRNA gene sequencing of pediatric fecal samples reveals the *Helicobacter pylori* status, crowding and the intake of traditional fermented cereal as parameters affecting microbiota diversity

To examine fecal microbiota composition and diversity relative to a variety of lifestyle factors in an urban African pediatric population, we took advantage of a birth cohort assembled between 2018 and 2022 in the only public hospital and two associated health centers of the Ethiopian city of Adama (Ethiopia's second largest city, population 600.000). Children included in the cohort were monitored from birth until age 6, and were annotated with numerous lifestyle conditions and choices, ranging from delivery mode to breastfeeding, nutritional status, housing conditions, family size, and prescribed medications. The *H. pylori* status of a subset of children was determined by serology. IgA coating of the gut microbes of a subset of children, for which sufficient material was available, was determined by flow cytometry. Two hundred seven 2–5-year-old children of the total cohort of 1256 children were randomly selected for 16S rRNA gene sequencing to determine the composition of the fecal microbiota; of these 207, 105 were female and 102 were male, and the mean age at the time of sampling was 3.7 years (11 were 2 years old, 62 were 3 years old, 114 were 4 years old and 19 were 5 years old). The cohort of 207 children was quite representative of the entire cohort with respect to the age at the time of sampling and the proportion of children born by cesarian section, but showed a more balanced representation of breast-fed and formula-fed children than the total cohort (Supplementary Data 1). Bacterial genomic DNA was of sufficient quality despite the logistical challenges associated with sampling (some of which happened on site, i.e., at the hospital of health center, and some at home) and intercontinental shipping. To assess the impact of metadata factors on overall gut microbiota structure, we first analyzed the 16S rRNA gene sequencing data using Bray–Curtis dissimilarities and PERMANOVA, with additional dispersion testing. None of the factors explained a significant proportion of the variance (Supplementary Data 2; note that the confounding factor of wasting was removed as it showed collinearity with age, Supplementary Fig. 1a). Also, unsupervised sample clustering based on the top 40 most abundant genera revealed no obvious segregation of the

children's microbiota based on their age, nutritional status and other parameters (Supplementary Fig. 1b). However, a closer examination of the α-diversity in relation to individual parameters uncovered several interesting patterns (Fig. 1a–c). Sex, breastfeeding, access to a toilet within the house or apartment, delivery mode (16% of children in our cohort were delivered by cesarean section) and family size all had no obvious effect on α-diversity as determined by Shannon index (Fig. 1a; Supplementary Fig. 1c, d and Supplementary Data 3); it should be noted however that average family sizes were quite small in our Adama population and only 12% of children in our cohort are being raised in families with six or more members. To investigate the effect of cesarean section in more detail, we divided the samples into bins based on their Shannon index; the bins with the lowest Shannon indices/lowest α-diversity had the highest fraction of children born via cesarean section and the bins with the highest Shannon indices had the lowest fraction (Supplementary Fig. 1e). Cesarean section prevalence showed no statistically significant differences across six equal-count Shannon bins after adjustment for age, and treating Shannon diversity as a continuous predictor yielded a similar conclusion (Supplementary Data 4). Other parameters had a much clearer effect on α-diversity; these were age, stunting, *H. pylori* infection, housing conditions and the intake of fermented cereal (Fig. 1b, c and Supplementary Data 3). α-diversity was higher in 3-year old children than in the 4- and 5-year old's, and, surprisingly, in children affected by stunting ($p = 0.019$; with stunting defined by the WHO as height-for-age that is more than two standard deviations below the median; Fig. 1b). Infection with the gastric pathobiont *H. pylori*, which in East Africa continues to be highly prevalent and which was serologically confirmed in 40% of examined children in our cohort (positive serology increasing with age; Supplementary Fig. 1f), was linked to lower α-diversity ($p = 0.019$; Fig. 1c). An important parameter affecting α-diversity was the number of rooms available per family, with a single (over multiple) room(s) being predictive of high diversity ($p = 0.002$; Fig. 1c). Interestingly, the regular consumption (at least once per week, and up to twice per day) of a locally grown cereal, teff (*Eragrostis tef*) in the form of pancakes (Injera) or traditional breads made from fermented dough, resulted in a higher diversity of the fecal microbiota ($p = 0.05$; Fig. 1c and Supplementary Fig. 1g). Higher α-diversity in *H. pylori*-negative and teff-consuming children, and children living in crowded conditions was attributable more to higher evenness than to a higher number of observed Amplicon sequence variants (ASVs), whereas stunting resulted in more observed ASVs (Supplementary Fig. 2a, b and Supplementary Data 3). Other factors would have been of interest, but could not be investigated; these include medicinal interventions and the effects of group B streptococcus (GBS) prophylaxis. The fraction of children that had been subjected to some form of medicinal intervention before their stool was sampled was so low in our cohort at 6% that no meaningful comparison was possible. GBS prophylaxis, i.e., the administration of antibiotics during labor for prevention of GBS transmission from mother to baby, is not performed in our participating health centers. We speculated that the level of IgA coating, i.e., the fraction of individual bacteria coated by IgA, would be associated with α-diversity, but this was not the case (Fig. 1d and Supplementary Fig. 2c). Interestingly, we were able to identify microbial indicators of high and also of low α-diversity. In particular, a high abundance of *Bifidobacterium* was indicative of low α-diversity; *Faecalibacterium*, *Ruminococcus*, *Anaerostipes* and several other bacteria were overrepresented in samples with high α-diversity (Fig. 1e; Supplementary Fig. 3a and Supplementary Data 5). We further asked whether any specific genera would be (positively or negatively) associated with the four parameters significantly affecting diversity in our cohort; this was however not the case, as no statistically significant associations were observed (Supplementary Fig. 3b). In conclusion, we find that *H. pylori* infection, a traditional diet based on fermented cereal, crowding/restricted living conditions, stunting and the presence of certain genera are all predictive of fecal microbiota diversity; in contrast, lifestyle choices and conditions, such as delivery mode, breastfeeding or family size have less of an impact on diversity in our cohort.

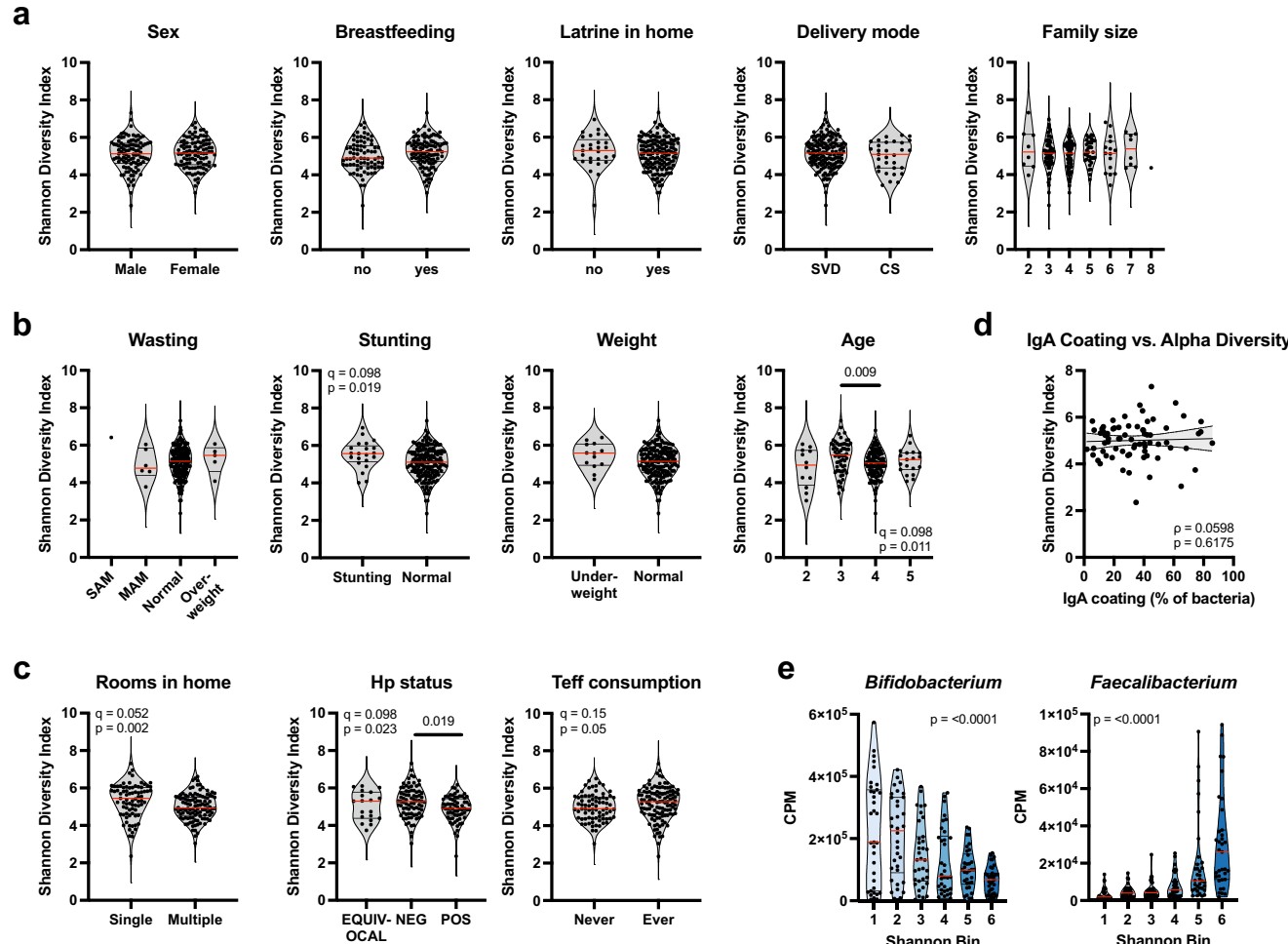

**Fig. 1 | Associations of Shannon diversity with demographic, lifestyle, immune, and microbial factors. a** Shannon diversity index stratified by categorical variables: sex, breastfeeding status, presence of a latrine in the home, delivery mode, and family size. **b** Shannon diversity index stratified by nutritional and demographic factors: wasting, stunting, weight-for-age, and chronological age. **c** Shannon diversity index stratified by household and lifestyle factors: number of rooms in the home, *Hp* infection status, and frequency of teff consumption. Statistical significance in (**a**–**c**) was assessed using Wilcoxon rank-sum or Kruskal–Wallis tests with post-hoc testing and BH correction where appropriate. Only results with $p \leq 0.05$ are displayed in the figure with their corresponding *q*-value ($n = 207$). **d** Correlation between Shannon diversity index and the proportion of IgA-coated bacteria measured by flow cytometry. Spearman's correlation coefficient ($\rho$) and *p*-value are reported; a simple linear regression line is shown with a 95% confidence band ($n = 72$). **e** Relative abundance (counts per million, CPM) of *Bifidobacterium* and *Faecalibacterium* are shown in relation to Shannon diversity bins. Associations were tested using linear regression models, including Shannon diversity and age. *p*-values from the regression models are plotted ($n = 207$). Red lines in (**a**, **b**, **c**, **e**) indicate medians.

## Metagenomic analyses reveal metabolic functions of previously unknown species-level genome bins associated with teff consumption

To gain more in-depth insights into the microbial species of our cohort, and the metabolic functions that might be associated with specific clinical and lifestyle factors, we selected 105 children (Supplementary Data 1) among the original cohort of 207 for metagenomic analysis (Supplementary Data 6). To assess whether community composition differed across host and environmental factors, we performed PERMANOVA analyses followed by tests for homogeneity of group dispersion. None of the tested variables explained a significant proportion of beta diversity variation (Supplementary Data 7). The factor most strongly associated with compositional shifts was frequency of teff consumption (PERMANOVA: $R^2 = 0.013$, $p = 0.111$; Supplementary Data 7). *H. pylori* infection status accounted for 3.1% of the variance but was not significant ($p = 0.28$). Other factors, including delivery mode, breastfeeding status at 18 months, stunting, age, household size, and the presence of a latrine or separate bedroom, explained ≤2.3% of the variance and were nonsignificant ($p > 0.1$). Tests for dispersion confirmed that none of the

associations were confounded by unequal within-group variance (all $p > 0.05$; Supplementary Data 7). A closer examination of the α-diversity in relation to individual parameters confirmed some of the patterns revealed by the 16S rRNA gene sequencing (Supplementary Data 8). Contigs resulting from shotgun metagenomic sequencing were binned into metagenome-assembled genomes (MAGs) (only bins with >90% completeness and <5% contamination were retained)[22]. MAGs were assigned to species-level genome bins (SGBs) using PhyloPhlAn, which was also employed to construct a phylogenetic tree (Fig. 2a). A total of 713 SGBs could be assigned, of which 587 represented known (kSGBs) and 126 represented unknown (uSGBs) SGBs. As already determined by 16S rRNA gene sequencing, Firmicutes contributed the vast majority of SGBs (73%), followed by Actinobacteria, Bacteroidota and Proteobacteria (Fig. 2a). Although teff-consuming children showed numerically higher carriage of uSGBs and kSGBs compared with nonconsumers (Fig. 2b), regression models did not support a statistically significant association (Supplementary Data 9). Teff consumption differed by age group also in the cohort of 105 metagenome-sequenced children; in particular, 5-year-olds were markedly less likely to consume teff

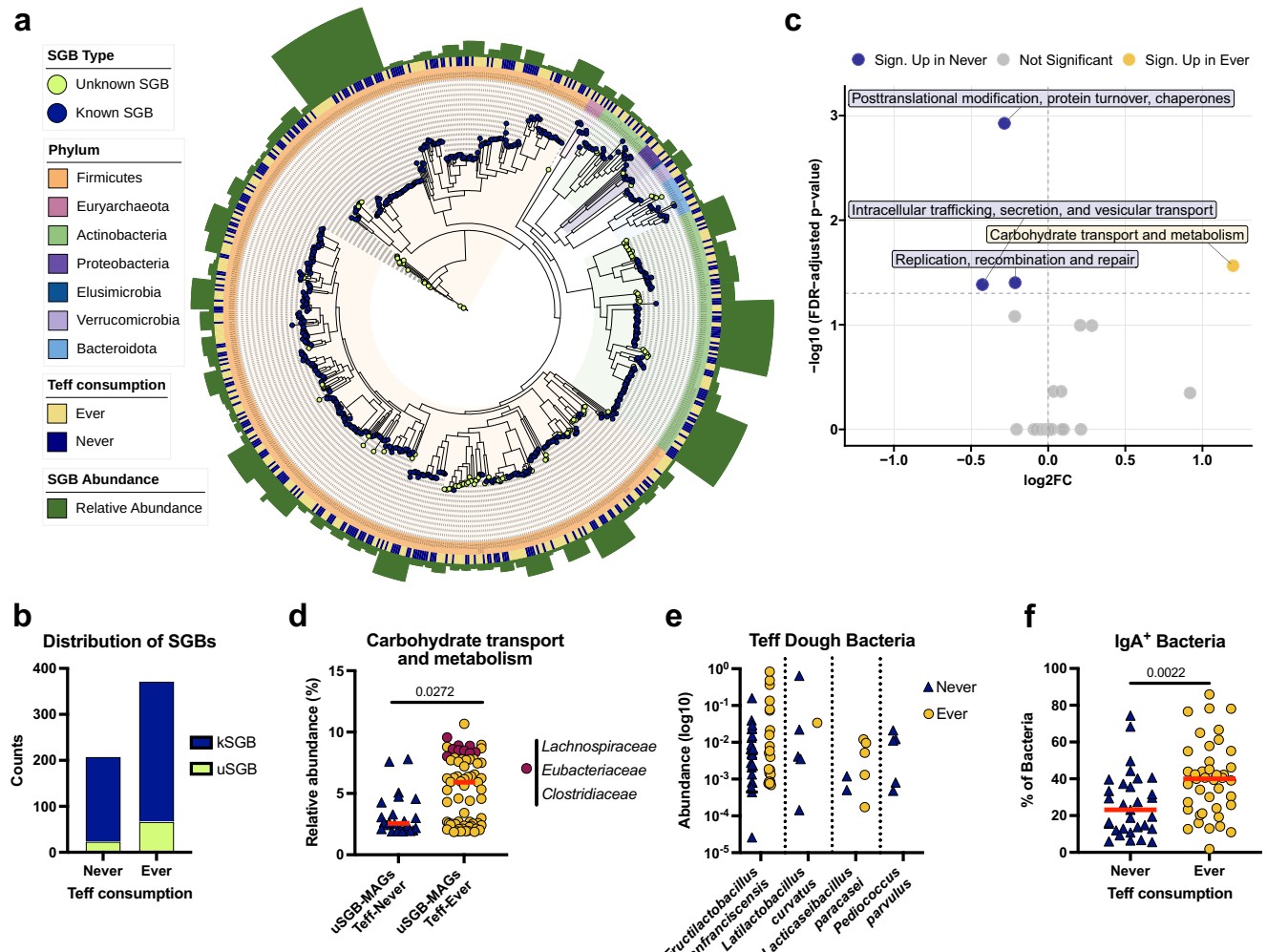

**Fig. 2 | Phylogeny and functional signatures of gut microbiota are shaped by diet in Ethiopian children. a** Phylogenetic tree of high-quality metagenome-assembled genomes (HQ MAGs) reconstructed from metagenomic sequencing data of urban Ethiopian children, generated using PhyloPhlAn. Tips are colored based on whether the assigned species genome bin (SGB) is known or unknown. Tree clades are colored in the background and in the innermost ring by phylum. The additional annotation rings displays the teff consumption metadata. The outer ring represents the relative abundance of each SGB across samples. (n = 105) **b** Distribution of known and unknown species-level genome bins in teff consumers versus non-consumers. Associations with teff consumption were evaluated using generalized linear models for presence/absence and negative binomial models for per-sample counts. No statistically significant associations were detected (Supplementary Data 9). **c** Volcano plot showing differential relative abundances of COG functional categories between teff consumption groups. The x-axis represents the log$_2$ fold change in median relative abundance, and the y-axis shows statistical significance as −log$_{10}$ of the FDR-adjusted *p*-value. Each point corresponds to a COG category,

colored by significance and direction. Categories significantly enriched in teff consumers (ever) are shown in yellow, those enriched in nonconsumers (never) in blue, and nonsignificant categories in gray. Labels are shown for categories with FDR < 0.05. Dashed lines indicate the significance threshold (FDR = 0.05) and zero effect size. **d** Column scatter plot of the relative abundance of the COG category carbohydrate transport and metabolism in uSGB-MAGs from teff-consuming and nonconsuming groups. In the teff-consuming group, uSGB-MAGs with a relative abundance >8% and an assigned family-level taxonomy are annotated in burgundy. Red lines indicate medians. Statistical significance was assessed using Wilcoxon rank-sum tests with BH correction (n = 91). **e** Relative abundance of bacteria previously identified in traditional teff dough, showing no significant difference between teff-consuming and nonconsuming children (n = 105). **f** Column scatter plot showing the percentage of IgA-coated bacteria in teff-consuming versus non-teff-consuming children. Red lines indicate medians. Statistical significance was assessed using Wilcoxon rank-sum tests (n = 72).

compared to 2–4-year-olds (odds ratio = 0.03, 95% CI 0.003–0.198; Fisher's exact test, *p* < 0.0001; Supplementary Data 10). Thus, this age group was excluded from subsequent age-stratified analyses.

Given that teff consumption was not significantly associated with the prevalence or number of uSGBs, and that no taxa showed differential abundance between teff consumers and nonconsumers, we next examined the functional potential of these uSGBs to determine whether teff consumption might instead be linked to differences at the metabolic or pathway level. We therefore functionally annotated the 67 uSGBs-MAGs found in teff consumers and the 24 uSGBs-MAGs found in the non-teff-consuming children with their clusters of orthologous genes (COG) categories; 22 COG categories were represented in the examined uSGBs-

MAGs, albeit at different relative abundances (Supplementary Fig. 4a). The COG category carbohydrate transport and metabolism was significantly enriched in teff-consumer uSGB-MAGs (Fig. 2c, d; Supplementary Fig. 4b–e and Supplementary Data 11), with highest relative abundances of this COG category found in newly sequenced genomes belonging to the *Lachnospiraceae*, *Clostridiaceae* and *Eubacteriaceae* families of bacteria. We looked for the presence of species previously found in fermented teff dough[13], and indeed were able to detect four species (*Lacticaseibacillus paracasei* SGB7142, *Fructilactobacillus sanfranciscensis* SGB7164, *Latilactobacillus curvatus* SGB7249 and *Pediococcus parvulus* SGB7180); their abundance was, however, roughly equal in teff-consuming and other children (Fig. 2e). Interestingly, teff-

consuming children were on average colonized by more IgA-coated bacteria than their non teff-consuming counterparts (Fig. 2f).

The available metagenomes further allowed us to search for and quantify antimicrobial resistance (AMR) genes and to correlate them with lifestyle factors and living conditions. The most commonly detected AMR genes coded for resistance to tetracycline, rifampicin and mupirocin (Supplementary Fig. 5a). Whereas the number of rooms in the family home, or the delivery mode was not correlated with the detection of AMR genes, we found numerically higher numbers of AMR genes in teff consumers, and in children who had access to a latrine in their (oftentimes single) living room; however, these associations were not statistically significant (Supplementary Fig. 5b and Supplementary Data 12). In summary, our metagenomic analyses have allowed us to describe and functionally annotate numerous previously unknown bacterial species, mainly belonging to the class of Clostridia, which colonize the GI tract of teff consumers and possibly contribute to the observed increased α-diversity of such traditionally nourished children.

### Metagenomic analyses reveal large differences in the fecal microbiota composition of rural and urban Ethiopians

Given the availability for metagenomic comparison of a published cohort of rural age-matched Ethiopian children (from the village of Gimbichu, about 60 km away from Adama)[19], we set out to compare the metagenomes of rural and urban Ethiopian children. Interestingly, non-Western indicator species identified not only in rural Ethiopians, but also in rural children from other parts of Africa, i.e., Ghana and Tanzania[19] were largely absent in our urban cohort. These included *Segatella copri*, *S. hominis* and *S. brasiliensis*, as well as *Prevotella* species, all of which were enriched in rural, but not urban Ethiopians (Fig. 3a, b). Other genera (*Blautia*, *Holdemannella*, and *Anaerobutyricum*) were strongly overrepresented in the urban children (Fig. 3a, b). Teff consumption did not significantly alter overall similarity to rural microbiomes ($p > 0.2$; Supplementary Data 13). The *Prevotella*/*Bacteroides* ratio, an accepted indicator of a diet rich in plant carbohydrates and fibers vs. a diet rich in animal fat and protein[23–25], was much higher in the rural than the urban children (Fig. 3c). Similarly, the Firmicutes/Bacteroidota ratio, an indicator of dietary sugar intake especially in the form of sugar-sweetened beverages[26] was higher in the urban children (Fig. 3c). Metabolic pathway analyses revealed numerous pathways to be enriched in the urban or the rural children (Fig. 3d and Supplementary Fig. 6a). Among the pathways enriched in the urban children were pathways required for the fermentation of the simple sugars glucose, lactose and galactose, such as the Bifidobacterium shunt, and pathways linked to xenobiotic detoxification and pollutant breakdown (Fig. 3d, e). Pathways typical of the rural children were enriched for degradation of plant-based glycosides (such as beta-D-glucuronoside, D-fructuronate, and hexuronate) and vitamin synthesis (Fig. 3d, f). We infer from the combined results that the urban and rural children differ with respect to their diet, which manifests in a substantially different gut species composition and likely has consequences for gut and overall health.

### The microbiome of urban Ethiopian children has more in common with European than with rural African children

Given the availability of fecal metagenomic data for a cohort of age-matched (2–5 year old) Italian children (Supplementary Fig. 7a)[13], we compared the three cohorts with respect to their most abundant genera and their dominant metabolic pathways, and the presence of AMR genes. Unsupervised clustering of genera with >15% mean relative abundance segregated urban and rural Ethiopian children, and Italian children onto distinct branches of the dendrogram tree (Supplementary Fig. 7b). The same was true when the children were clustered based on the top 20 most abundant genera per cohort, resulting in 46 genera in total (Supplementary Fig. 7b, c). Rural signature genera, such as *Segatella* and *Prevotella* were absent not only in urban Ethiopian, but also in Italian children (Supplementary Fig. 7b, c). Conversely, the top urban genera *Blautia* and *Bifidobacterium* were shared with Italian children (Supplementary Fig. 7b, c). Interestingly, several Italian

signature genera—*Roseburia*, *Bacteroides*, *Phocaeicola*, *Alistipes*—were absent in all African children, and there was even one genus, *Faecalibacterium*, which rural, but not urban Ethiopians shared with Italian children (Supplementary Fig. 7b, c). Several *Treponema* species associated with traditional lifestyles[16,27] did not make the top 20 genera cutoff, but were present in rural Ethiopians, and not the other two cohorts (Supplementary Fig. 8a).

The three cohorts were highly divergent as judged by Bray–Curtis dissimilarity index, both when calculated based on the microbiota species composition (Fig. 4a) and when calculated based on metabolic pathways (Fig. 4b). Hierarchical clustering of the top 30 metabolic pathways per cohort confirmed the substantial differences between the three cohorts in terms of their main microbial metabolic activities, with urban Ethiopians seemingly bearing more resemblance to Italian children than to their rural counterparts (Supplementary Fig. 8b). Furthermore, whereas 20% of urban Ethiopian SGBs (49 of 243) were shared with the Italian children's SGBs, only 4.5% (11 of 243) were shared with rural Ethiopian SGBs (Fig. 4c). The same pattern was observed when only uSGBs were taken into consideration (Fig. 4c). In order to quantify and compare the presence of AMR genes in the three cohorts, we mined the metagenomes of the Italian and rural Ethiopian cohorts for AMR genes. Rural Ethiopian metagenomes had on average the fewest AMR genes, and urban Ethiopians the highest number, with Italian children falling in between (although the differences were not significant; Fig. 4d). Tetracycline resistance was the most commonly detected AMR, followed by rifampicin and mupirocin resistance (Fig. 4e and Supplementary Fig. 8c). Italian and urban Ethiopian metagenomes showed similar AMR drug class profiles, which in turn differed from those found in rural Ethiopian metagenomes (Fig. 4e). These AMR gene and drug class profiles represent an additional dimension along which the three cohorts differ. Finally, we mined the metagenomes for the (differential) presence of putative bacterial pathogens; indeed, the prevalence of *E. coli*, *Enterococcus*, *Klebsiella pneumoniae* and *Clostridium perfringens* was generally higher in urban Ethiopians than in their rural counterparts, or in Italian children (Supplementary Fig. 8d). The combined results suggest that urban Ethiopian children have adopted a westernized microbiome that is characterized by loss of traditional African signature microbes, a metabolic switch to the utilization of simple refined and dairy sugars, and the acquisition of a diverse set of AMR genes.

## Discussion

We present here the results from microbiome analyses conducted on an African pediatric cohort that differs from previously examined cohorts in that the children are raised in an urban environment in comparatively small families, with relatively normal growth trajectories (6% of children were underweight and 10% exhibited stunting) and ready access to health care. Around 15% of children were born via C-section, a rate that is half that of Western countries. None of the children were exposed to intrapartum GBS prophylaxis, as this practice is not part of routine perinatal care in Ethiopia. We were able to examine microbiome features relative to >10 health, developmental, nutritional and lifestyle parameters that were collected through extensive parental questionnaire-based reporting. This analysis confirmed several known or plausible predictors of fecal microbial diversity, such as breastfeeding (although only a trend was seen), crowding (the whole family living in one, rather than in multiple rooms) and nutritional restriction that is evident as stunted growth and being underweight. Stunting, or linear growth failure, affects 22% of children worldwide[28]. It has been associated with immaturity of gut microbiota, i.e., a delay in the assembly of an age-typical microbial community, but also with higher levels of pathogenic and inflammogenic taxa and lower levels of beneficial taxa, such as *Bifidobacterium*[29–31]. Remarkably, severe caloric restriction and the associated rapid weight loss was shown to result in a temporary increase in α-diversity in an interventional trial, with microbiota diversity returning to baseline once the caloric restriction had terminated[32]. Microbiota transplantation of post-diet samples harvested from the trial patients to recipient mice decreased their body weight and adiposity relative to pre-diet

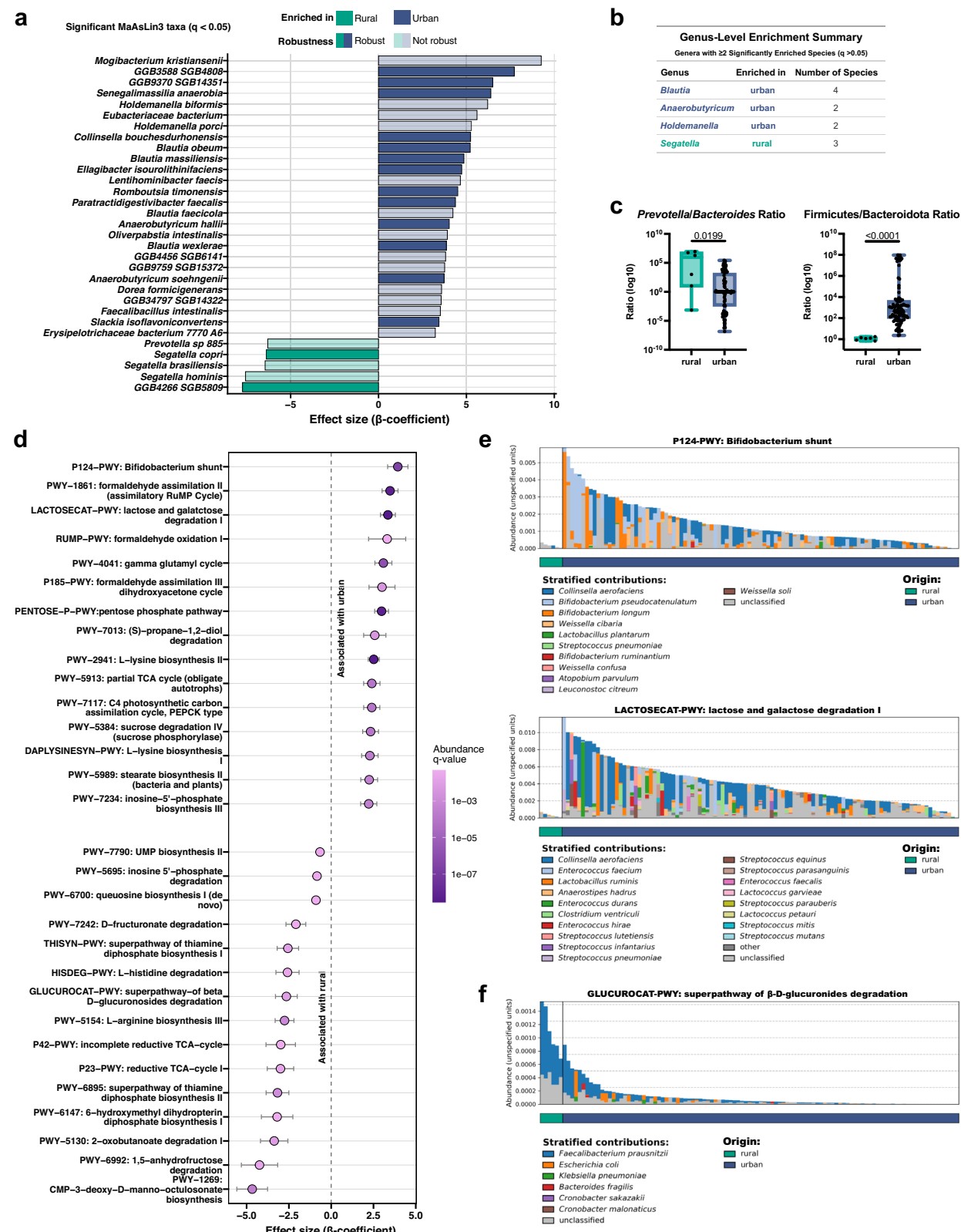

microbiota recipients[32]. Indeed, dysbiosis in stunted children has been proposed to be both cause and consequence of the condition, with stunting altering the gut microbial community and with the resulting dysbiosis exacerbating the condition through impaired nutrient absorption, chronic inflammation, altered short-chain fatty acid production, and perturbed hormonal and signaling pathways[33].

Other parameters, i.e., the birth mode and family size, were not correlated with microbiome features. This was perhaps not surprising given the mean (advanced) age of 3.7 years of our cohort. Previous studies have documented a strong effect of C-section delivery on the neonatal (days 4, 7, and 21 post delivery) microbiota that was much less evident by 9 months of age[7], although a longitudinal study conducted until the age of 2 years

**Fig. 3 | Urban and rural Ethiopian children differ with respect to their gut microbiota composition and metabolic functions. a** Diverging plot of species-level abundance differences between urban ($n = 105$) and rural Ethiopian ($n = 6$) children. Bars show MaAsLin3 β-coefficients for species significant in the full model ($q < 0.05$), adjusted for centered age. Positive values indicate higher abundance in urban samples, and negative values indicate higher abundance in rural samples. Opacity encodes robustness status from the resampling analysis (25 urban vs. 6 rural per iteration; 100 iterations): robust (criterion met: significant in ≥25% of iterations with ≥75% consistent effect sign) at full opacity, not robust at reduced opacity. Species are ordered by effect size. **b** Summary table showing genera enriched in either the urban or rural cohort, along with the number of enriched species per genus, corresponding to (**a**). **c** Health-related microbial ratios. Log$_{10}$-transformed abundance ratios of *Prevotella*-to-*Bacteroides* (genus level) and Firmicutes-to-Bacteroidota (phylum level) plotted by origin. Statistical significance was assessed using the Wilcoxon rank-sum test. Box-and-whisker plots show median, interquartile range, and min-max values. **d** Coefficient plot of pathway abundance associations, displaying the 15 significantly different pathways with the highest and lowest β-coefficients as identified by MaAsLin3, indicating enrichment in either urban or rural children. Corresponding FDR-adjusted *p*-values and effect sizes (β-coefficients) are plotted. **e** Bar plots of stratified abundance of the P124-PWY (Bifidobacterium shunt) and LACTOSECAT-PWY (lactose and galactose degradation I) pathways, split by origin. Example of two pathways positively associated with urban children as identified by MaAsLin3. **f** Bar plot of stratified abundance of the GLUCUROCAT-PWY (superpathway of beta-D-glucuronosides degradation), split by origin. Example of a pathway positively associated with urban children as identified by MaAsLin3. All subpanels include urban $n = 105$ and rural $n = 6$ samples.

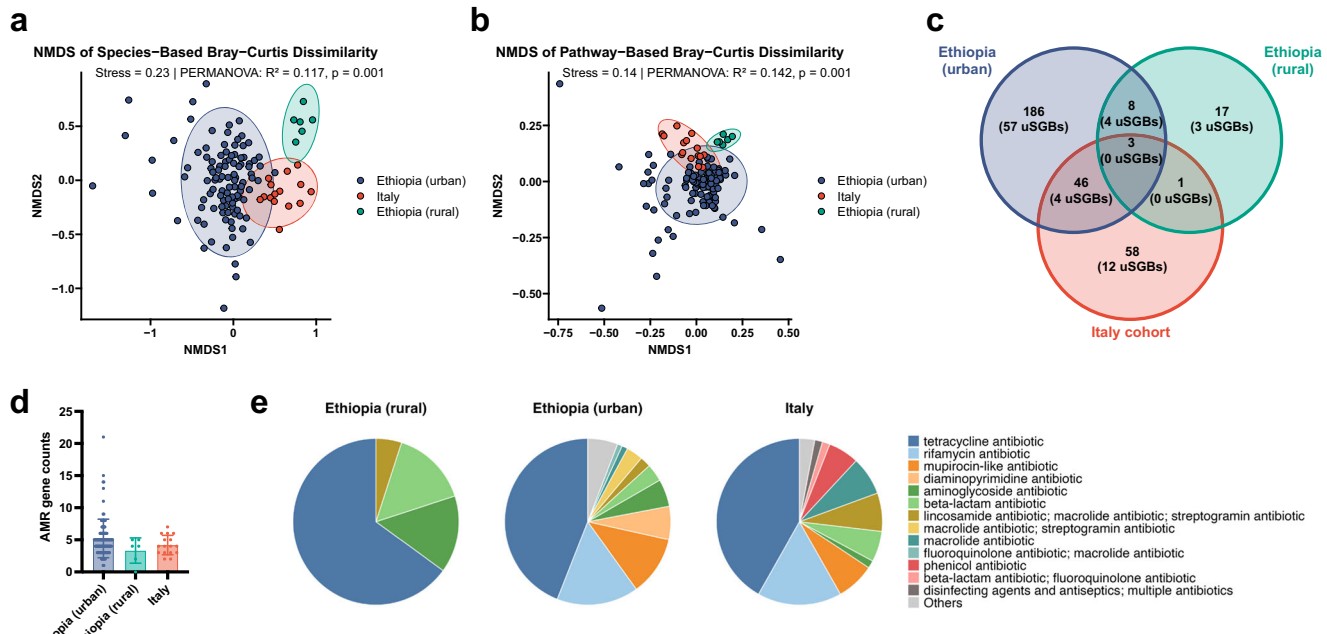

**Fig. 4 | Urban Ethiopian children are more similar to European children than to their rural Ethiopian counterparts. a** Non-metric multidimensional scaling (NMDS) plot based on Bray–Curtis dissimilarity of species profiles. **b** NMDS plot based on Bray–Curtis dissimilarity of pathway profiles. Stress value, explained variance ($R^2$), and significance ($p$) from PERMANOVA are indicated in (**a, b**). Samples are colored by cohort (urban Ethiopian, rural Ethiopian, and Italian). **c** Venn diagram showing the overlap of species genome bins (SGBs) identified using PhyloPhlAn among urban Ethiopian, rural Ethiopian, and Italian cohorts. Numbers in brackets indicate the number of unknown SGBs (uSGBs) within each group. **d** Column scatter plot of AMR gene counts across the three cohorts, revealing no significant differences as assessed by the Kruskal–Wallis test with post-hoc testing. Means ± SD are shown. **e** Pie charts of the top 10 antimicrobial resistance (AMR) drug classes per cohort; drug classes with fewer counts are grouped as "others." ($n = 105$ urban Ethiopian, $n = 6$ rural Ethiopian, $n = 16$ Italian).

documented a persistent negative effect of C-section delivery on α-diversity up until that age[6]. We identified two additional factors impacting fecal microbiota diversity. One was the *H. pylori* status; *H. pylori*-infected children had a lower α-diversity than their non-infected counterparts. While the literature on the effects of *H. pylori* on the gastric microbiota is quite consistent, with a reduced microbial diversity and abundance observed in *H. pylori*-infected relative to control children due to *H. pylori*'s numerical predominance[34,35], the relationship of *H. pylori* presence with intestinal/fecal microbiota diversity is more complicated. Several studies, conducted on both adults[36,37] and children[38], showed a higher fecal diversity in the infected individuals; others found no differences or, like our study, a decreased evenness of species[39–41]. Other parameters, such as socioeconomic status, gastritis (infection-driven or not) and linear height and weight gain, appeared as more important determinants of gut microbiota diversity in the examined cohorts than the *H. pylori* status[40,41].

The other determinant affecting microbiota diversity in our cohort was the consumption of teff, typically in the form of fermented bread products. Teff-consuming children showed a higher fecal α-diversity and more SGBs (known and unknown) than their non-teff-consuming counterparts. A higher proportion of their fecal bacteria was coated with IgA. IgA coating is generally believed to be associated with gut health, and overall health[42]; children with allergies, for example, harbor lower proportions of IgA-coated bacteria once they have developed symptoms[43], and in the years preceding the development of overt disease[44]. Our levels of IgA coating ($34 \pm 20\%$ of all fecal bacteria) is in the range of on average 10–36% reported by others[45–48]. Many of the uSGBs enriched in teff-consuming children could be assigned to bacterial families—*Lachnospiraceae*, *Eubacteriaceae*, and *Clostridiaceae*—that are all known to produce short-chain fatty acids from the fermentation of diverse plant polysaccharides via carbohydrate-active enzymes[49]. The increased gut microbiota diversity of teff-consuming children was also observed in the study of rural Ethiopians to whose metagenomic profiles we had access[13,19]. In these children, the increased gut microbiota diversity could at least partly be linked to the presence of bacteria found in teff products[13]; we were able to detect the same four main species in our cohort, but could not find evidence for their overrepresentation in the teff-consuming children relative to the children who had never consumed teff products.

The comparison of metagenomic profiles of our urban Ethiopian children with those of an age-matched rural Ethiopian cohort and those of an Italian cohort revealed numerous interesting differences and similarities. Signature species of a traditional African lifestyle, such as *Segatella* and certain *Prevotella* species[13,19], were entirely absent in the urban Ethiopians and Italians. Enterotype-based indicators, such as the *Prevotella/Bacteroides* and Firmicutes/Bacteroidota ratios that are tightly associated with diet and lifestyle[23–26], reveal strong differences between rural and urban Ethiopians. Numerous metabolic pathways, of which some are attributable to the degradation and utilization of plant-derived polysaccharides, are enriched in rural Ethiopians but absent in urban Ethiopians; the urban children's microbiota is rather selected for the utilization of the simple sugars glucose, lactose and galactose. Our results are well in line with numerous previous studies showing that industrialized populations have microbiotas that are dominated by *Bacteroidaceae*, whereas traditional populations across African, Asian, and South American continents, which include a range of lifestyles from rural agriculturalists to hunter-gatherers, have microbiotas that are distinguished by their abundances of Prevotellaceae, both families being members of the Bacteroidetes phylum[14,20,50].

The mining of metagenomic data for AMR genes allowed us to quantitatively annotate the gene counts on the one hand, and on the other hand, assign drug classes corresponding to these genes. Whereas the gene counts did not differ significantly between rural and urban populations, the diversity of drug classes clearly differentiated urban and rural cohorts. This finding is in line with published reports from other geographical areas[51,52].

Differences in AMR profiles are often shaped by differences in antibiotic exposure and healthcare, and it is plausible that such factors contribute to the patterns observed here[53]. However, comparable metadata on antibiotic use, treatment history, or healthcare utilization were not available for the cohorts, preventing a direct assessment of these potential drivers. Within these constraints, the AMR gene repertoire nevertheless provides an additional dimension along which the analyzed populations diverge.

While the analyses yield consistent and biologically plausible patterns, several practical considerations should be noted. Sample sizes differed across the urban, rural, and Italian cohorts, and the rural metagenomic subset was comparatively small, which may limit the detection of more subtle differences. In addition, the rural samples did not fully match the age range of the urban cohort, and feeding information was not available across all cohorts. These factors mainly affect cross-cohort comparisons, whereas the intra-cohort associations were examined within a well-powered and well-characterized data set. We therefore interpret between-cohort contrasts cautiously and regard them as exploratory patterns that can guide future, more targeted investigations.

Our results indicate that diet and lifestyle, rather than geography per se, dictate the composition of an African urban intestinal microbiome; the changes in microbiota composition reflect the adoption of a Western lifestyle and diet that likely has consequences for gut and metabolic health, and overall health in the long run.

## Methods
### Cohort description and sample collection
The birth cohort study from which the samples were derived was conducted in Adama city, Ethiopia, from April 2018 to December 2022[54,55]. The study was carried out at one public hospital and two primary health centers. Most participants were urban residents with relatively better access to utilities, engaging mainly in small trade. The local diet centers on teff-based foods, such as injera, while wheat-based products and animal products are also increasingly being consumed. One thousand two hundred fifty-six mother-child pairs were enrolled and followed for 5 years. Baseline data were collected during antenatal care, with follow-up assessments at birth and at 9, 18, 24, 36, 48, and 60 months. The study examined tuberculosis infections, pregnancy outcomes, child growth and nutrition and autoimmune diseases. Group B Streptococci (GBS) prophylaxis, i.e., the administration of antibiotics during labor for prevention of GBS transmission from mother to baby, is not performed at our health centers. Complementary feeding practices were recorded: 84% of children received complementary foods around 6 months (with 5.9% starting earlier), often while still breastfeeding. Early feeding sometimes included cow milk-based formula (initiated before 2 months), and common foods introduced by 6 months were vegetables, fruits, cereals, and legumes, with meat or eggs typically added after 6 months. Homemade solid foods generally relied on locally available cereals and starchy tubers. The protocol for the birth cohort study conducted in Adama was approved by the National Ethical Review Board of Ethiopia (ref. no. 3.10/16/2018). Written informed consent was obtained at the initial mother-child recruitment. Fecal samples were collected from healthy children during birth cohort follow-up. Immediately after collection, samples were stored at −80 °C without the addition of preservatives until further processing. In addition to fecal samples, further clinical and familial data were collected, including patient demographics and health information. *H. pylori* status was determined of 752 children by IgG serology, using the ELISA kit RE56381 (Tecan). From these, 207 children were randomly selected, and their fecal samples were shipped on dry ice to Switzerland for microbiome analysis.

### DNA extraction
DNA was extracted from fecal samples (50 to 200 mg) using the DNeasy® PowerSoil® Pro Kit (#47016, QIAGEN). Samples were homogenized in the provided bead tubes by horizontal vortexing for 15 min, followed by processing according to the manufacturer's protocol. DNA concentration was determined using a NanoDrop spectrophotometer, and only extracts with an A260/280 ratio of ~1.8 and a concentration above 5 ng/μL were considered for downstream analyses.

### 16S rRNA gene sequencing
Extracted DNA was submitted to NovoGene (UK) for sequencing. 16S rRNA gene amplification targeted the V3-V4 hypervariable regions using primers 341F and 806R. Paired-end sequencing (250 bp reads) was performed on an Illumina platform, targeting an average sequencing depth of ~30,000 reads per sample. Raw sequencing reads were inspected for quality using FastQC[56]. Reads were trimmed and filtered using the cutadapt plugin in QIIME2[57] (version 2025.7). ASVs were identified with DADA2[58], and taxonomy was assigned using QIIME2's feature-classifier classify-sklearn, trained on the SILVA 138 rRNA reference database[59–61]. Alpha and beta diversity metrics were computed in QIIME2 to assess microbial community composition and variability after rarefying to 24000 reads.

### Metagenome sequencing
A subset of samples was selected based on age (2, 3, 4, and 5 years), with samples from 3 to 4-year-olds further selected based on *H. pylori* status and metadata completeness, resulting in a final set of 105 samples. Extracted DNA was submitted to NovoGene (UK) for sequencing. DNA libraries were prepared and sequenced on an Illumina NovaSeq X Plus platform, generating 150 bp paired-end reads (PE150). Raw sequencing reads were inspected for quality using FastQC[56] and summarized with MultiQC[62]. Reads were trimmed and quality filtered using Trimmomatic[63] (V.0.39). Additionally, reads were filtered to remove host contamination by alignment with the human genome (GRCh38) using Bowtie2[64] (v2.5.3) using the —fast-local parameter. After reads that mapped to the host were discarded, the non-host reads were then assembled into contigs using SPAdes genome assembler (metaSPAdes mode v3.15.5)[65]. Contigs were binned into MAGs with Metabat2[66] (v 2:2.15), and the quality of each bin was assessed using CheckM[67] (v1.2.2); only bins with >90% completeness and <5% human contamination (high-quality bins) were retained[21]. Taxonomic profiles were generated using MetaPhlAn4[68] (v 4.1.0, database vJun23_202403), and functional potential was characterized with HUMAnN3[69] (v 3.9, databases ChocoPhlAn v201901_v31 and UniRef90 v201901b). MAGs were assigned to species-level genome bins (SGBs) via PhyloPhlAn[70] (v3.1.68), which was also employed to reconstruct the phylogeny. The interactive tree of life[71] (iTOL, v7.2) was used to visualize the phylogenetic tree. Finally, the MAGs were functionally annotated using prodigal[72] (v 2.6.3) and DIAMOND[73]

(V2.0.11) in eggNOG mapper[74] (emapper-2.1.12) with the eggnog DB (V.6.0.). Antibiotic resistomes were identified using the resistance gene Identifier package[75] (RGI, v6.0.0) using the CARD database (v 3.2.6) implemented in the AMR pipeline (v1.2.4). All 16S rRNA and metagenomic data reported in this work are publicly available from the NCBI BioProject database (BioProject: PRJNA1345963).

### Integration of external metagenomic datasets

For the Italian and rural Ethiopian cohorts, we used publicly available raw sequencing data[13,19]. DNA extraction and library preparation for these datasets had been performed with the same DNeasy® PowerSoil® Pro Kit (#47016, QIAGEN) and NexteraXT DNA Library Preparation Kit (Illumina), followed by sequencing on an Illumina HiSeq2500 platform with 100 nt paired-end reads at a target depth of ~5 Gb/sample. In contrast, our newly generated Ethiopian cohort was sequenced on an Illumina NovaSeq X Plus platform with 150 nt paired-end reads at a target depth of ~12 Gb/sample. To ensure comparability, all raw reads from the external cohorts and our own dataset were re-processed through an identical bioinformatics pipeline (quality control, host filtering, taxonomic, and functional profiling, assembly, binning, and annotation). This harmonization mitigates potential platform-specific effects and enables valid cross-cohort comparisons.

### Flow cytometry staining of fecal samples

The protocol was modified from Conrey et al.[45]. Briefly, fecal samples were homogenized by pipetting, diluted in 1 mL PBS and kept on ice. Samples were centrifuged at $50 \times g$ for 15 min at 4 °C to remove larger particles. From the resulting supernatant, 100 μL were transferred into a 96-well deep-well plate and washed twice with PBS ($8000 \times g$ for 5 min at 4 °C). The optical density (OD) at 600 nm was measured and adjusted to OD = 0.1, using 200 μL of the adjusted suspension for further processing. Cells were washed once with 1 mL PBS supplemented with 1% bovine serum albumin (BSA) (PBS/BSA) ($8000 \times g$ for 5 min at 4 °C), and the supernatant was carefully aspirated. Samples were fixed in 100 μL 4% paraformaldehyde (PFA) for 20 min at room temperature (RT), followed by another wash with 1 mL PBS/BSA under the same conditions. After fixation, cells were resuspended in 100 μL blocking buffer (PBS supplemented with 10% normal goat serum) and incubated for 30 min on ice. Following a wash step with PBS/BSA, cells were incubated with 100 μL PBS/BSA containing anti-human IgA antibody (1:100, Alexa Fluor 647, Jackson ImmunoResearch, # 109-605-011) for 30 min on ice. After another wash with PBS/BSA, cells were stained with 100 μL of PBS/BSA containing SYTO BC (1:2000, Invitrogen, #S34855) for 15 min at RT in the dark. Finally, cells were washed (PBS/BSA), centrifuged, and resuspended in PBS/BSA for analysis. Control samples included unstained cells, single-stained controls for IgA and SYTO BC, and fluorescence minus one control for IgA.

### Statistical analysis

**Confounding factors.** To avoid collinearity between confounding factors, we assessed pairwise associations between categorical variables using Cramer's V statistic. Associations were visualized in a heatmap (Supplementary Fig. 1a). Variables with high correlation (Cramer's $V > 0.7$) were considered redundant, and only one representative variable was retained. "Wasting" was strongly correlated with "age (years)" and therefore excluded. The final set of categorical confounders included "Hp status", "teff consumption", "breastfeeding", "delivery mode", "age (years)", "stunting", "family size", "latrine in home", and "rooms in home".

To assess the impact of teff consumption and potential confounding factors on overall gut microbiota structure, we first analyzed the 16S rRNA gene sequencing data. Bray–Curtis dissimilarities were obtained from the qiime2 core-metrics pipeline, and permutational multivariate analysis of variance (PERMANOVA) was performed using the function "adonis2" from the vegan: community ecology package[76] (v 2.6–10) with 999 permutations. Each categorical factor (Hp status, breastfeeding, delivery mode, age (years), stunting, family size, latrine in home, number of rooms in home, and teff consumption) was tested individually. The proportion of explained

variance ($R^2$) and significance were recorded (Supplementary Data 2). Because PERMANOVA assumes equal group dispersion, we additionally tested this assumption using the function "betadisper" (followed by "permutest") from the vegan package, again with 999 permutations. Both ANOVA-based and permutation-based $p$-values were extracted (Supplementary Data 2).

The same analysis workflow was then applied to the shotgun metagenomic dataset, using Bray–Curtis dissimilarities derived from species-level MetaPhlAn4 profiles, calculated with the function "vegdist" from the vegan package. Results of PERMANOVA and dispersion testing for all factors are summarized in Supplementary Data 7.

### Alpha diversity for 16S rRNA gene sequencing

Alpha-diversity values were imported from the core-metrics output generated with QIIME 2.

To test for associations between α-diversity and metadata variables, we applied nonparametric tests depending on the number of levels per factor. Two-level variables were assessed using Wilcoxon rank-sum tests, and variables with more than two levels were tested using Kruskal–Wallis tests. Where applicable, post-hoc comparisons were performed: pairwise Wilcoxon rank-sum tests for two-level factors and Dunn's tests for multi-level factors. Within each factor, post-hoc $p$-values were adjusted using the Benjamini–Hochberg (BH) procedure.

Prior to testing, empty strings in metadata variables were treated as missing values and excluded on a per-test basis. Results are reported in Supplementary Data 3.

Associations between microbial diversity and delivery mode were assessed in children aged 2–5 years with available delivery information. Alpha diversity (Shannon index) was taken from the Qiime2 output. Shannon indices were divided into six equal-count bins using the "cut_number" function from ggplot2 package[77] in R, and the proportion of C-section births was calculated within each bin and age group. Logistic regression models with delivery mode (C-section vs. spontaneous vaginal delivery) as the outcome and age as a covariate were then fitted. In the binned model, Shannon diversity was included as a categorical predictor (six bins, lowest bin as reference), with odds ratios (ORs) and 95% confidence intervals (CIs) estimated for each bin and a likelihood ratio $\chi^2$ test used to assess the global association. In a second model, Shannon diversity was analyzed as a continuous predictor, yielding an age-adjusted ORs per unit increase in Shannon index with Wald test $p$-value. All results are reported in Supplementary Data 4.

ASV tables and taxonomic assignments were imported from QIIME 2 artifacts. Raw counts were converted to counts per million (CPM) using edgeR[78] after adding a pseudocount of 1 to avoid zeros. Target taxa were defined at the genus level. For each target taxon, CPM values were summed across all ASVs annotated to that taxon, yielding one CPM value per sample and taxon. Shannon values were divided into six bins (as describes above, labeled 1–6 from lowest to highest). A targeted set of genera (*Bifidobacterium*, *Faecalibacterium*, *Ruminococcus*, *Agathobacter*, *Anaerostipes*, *[Ruminococcus] gauvrenauii group*, and *Fusicatenibacter*) was selected based on significant associations with diversity in exploratory analyses, high abundance and variability (top 20 genera), and established roles in SCFA-related gut ecology. For each bacterial taxon, we used a linear regression model to test whether its CPM abundance was associated with alpha diversity and age. Shannon diversity was included as the 6 bins, and age as a continuous variable. From each model, we extracted the effect of age, including the estimate, standard error, and $p$-value, as well as the overall effect of Shannon diversity, which was tested with an F-test across all six categories (Supplementary Data 5). In addition, results from an analogous set of models using continuous Shannon values (instead of bins), as well as a sensitivity analysis repeating the bin-based model using 4–8 bins, are provided (Supplementary Data 5).

The Shannon diversity index was plotted against IgA coating (frequency of bacteria). Association was assessed using Spearman's rank correlation (ρ) with the corresponding $p$-value, computed in GraphPad Prism.

For visualization, a simple linear regression line with 95% confidence band was overlaid on the scatter plot.

ASV count data were aggregated at the genus level using SILVA taxonomy, and the 40 most abundant genera across all samples were selected. Counts were transformed to relative abundances per sample, expressed as percentages, and log-transformed with a small pseudocount. The resulting matrix was visualized as a heatmap using the ComplexHeatmap package in R, with samples split by Shannon diversity bins and genus-level taxonomy shown as row annotations. Study metadata, including diversity indices and clinical or household factors, were added as top annotations.

Associations between genus-level microbial abundances and host metadata were tested using MaAsLin3[79], which fits multivariable regression models for each microbial feature while controlling for covariates. Genus-level abundance tables were obtained from collapsed ASV counts and filtered to include only genera with non-zero variance. For each analysis, we tested the effect of the variable of interest (*H. pylori* status, stunting, teff consumption, and rooms in home) on microbial abundance while controlling for age as a continuous covariate (mean-centered, Age_c). Mean centering was applied to improve model stability and interpretability of the coefficients. All models were run with total-sum scaling (TSS) normalization and log transformation of microbial abundances, as implemented in MaAsLin3. Statistical significance was defined at FDR < 0.05 using MaAsLin3's BH-adjusted term-level *q*-values. No significant associations were detected in any of the models. To visualize effect size distributions, we therefore plotted the genera with the 10 highest and 10 lowest β-coefficients from each analysis, together with 95% CIs and corresponding FDR-adjusted values.

### Alpha diversity for metagenomics
Species-level relative abundance profiles were obtained from MetaPhlAn outputs. Rows annotated at the species level (containing "|s__") were extracted, while features resolved further to the strain level (containing "|t__") were excluded.

Alpha-diversity was quantified using the Shannon and Simpson indices (calculated with the function "diversity" of the vegan package), and observed richness was defined as the number of species with non-zero abundance per sample.

To test for associations between α-diversity and metadata variables, we applied nonparametric tests depending on the number of levels per factor, with post-hoc tests and adjustment were applicable as described above for the 16S rRNA gene sequencing data. Results are reported in Supplementary Data 8.

### SGB analysis
**uSGB and kSGB prevalence.** To investigate whether the presence of unknown (uSGB) or known (kSGB) species-level genome bins was associated with teff consumption, we generated binary indicators for each sample reflecting the presence (≥1) or absence (0) of uSGBs or kSGBs. Associations with teff consumption (ever vs. never) were tested using generalized linear models (GLMs) (via the "glm" function of R package stats). ORs with 95% CIs and *p* values are reported in Supplementary Data 9.

To assess whether the number of uSGB or kSGB species-level genome bins differed by teff consumption, we calculated per-sample counts of distinct uSGBs and kSGBs. Associations with teff consumption (ever vs. never) were evaluated using GLMs for count data, specifying a negative binomial distribution (implemented via the "glm" and "glm.nb" functions of the R packages stats and MASS [80]). Model coefficients for teff consumption were exponentiated to obtain rate ratios (RRs) with 95% Wald CIs, with corresponding *p* values reported in Supplementary Data 9.

To assess whether teff consumption frequency differed by age, we compared age groups using Fisher's exact test. Because only one child in the 5-year-old group reported teff consumption, we additionally performed a 2 × 2 Fisher's exact test contrasting 5-year-olds with all younger children (2–4 years). Effect sizes are reported as odds ratios with 95% CIs in Supplementary Data 10.

**COG.** To identify features differing in relative abundance between teff consumers and nonconsumers, we compared distributions using the Wilcoxon rank-sum test. For each feature present in both groups, a *p*-value was obtained and adjusted for multiple testing using the BH false discovery rate (FDR). COG categories with FDR < 0.05 were considered significant (Supplementary Data 11).

For visualization, we computed the $\log_2$ fold change in median relative abundance between consumers and nonconsumers. Volcano plots were generated with $\log_2$ fold change on the x-axis and $-\log_{10}$(FDR-adjusted *p*) on the y-axis. Significant categories were highlighted based on FDR only (no fold-change cutoff was applied).

### Comparison Ethiopia (urban) and Ethiopia (rural)
Species-level relative abundance tables were filtered to retain only features resolved to species while excluding features that were only resolved to higher taxonomic levels or to the strain level.

Microbiome-metadata associations were tested using MaAsLin3[79].

To account for significant differences in age distributions between cohorts (Supplementary Fig. 7a), age was included as a continuous covariate (mean-centered, Age_c) in all models. Mean centering was applied to improve model stability and interpretability of the coefficients.

All models were run with TSS normalization and log transformation of microbial abundances. As described before, significance was defined at FDR < 0.05 using MaAsLin3's BH-adjusted term-level *q*-values (qval_individual).

For the genus-level summary, we then retained species significantly associated with origin (qval_individual <0.05), and reported only genera with ≥ 2 significant species.

To assess robustness, we performed a resampling stability analysis. In each of 100 iterations, we included all 6 rural samples and a random subset of 25 urban samples, re-fitting the same MaAsLin3 model. For each contrast, we recorded the proportion of iterations with *q* < 0.05 (stability of significance) and the proportion of iterations in which the effect sign was the same (directional consistency). A species was considered "robust" if it reached significance in ≥25% of iterations and exhibited a consistent effect direction in ≥75% of iterations. For visualization, significant species from the full model (*q* < 0.05) were displayed, and robustness was indicated by bar opacity.

For pathway analysis, we applied Maaslin3 using age as a continuous covariate, as described above. The 15 significant pathways with the highest and lowest β-coefficients for abundance in relation to origin were plotted.

To test whether teff consumption influenced overall similarity to rural microbiomes, we calculated the Bray–Curtis distance of each sample to the rural centroid based on species-level relative abundances. We then fitted linear models with distance as the outcome and teff consumption (ever/never), age, and origin as predictors. To directly address whether teff consumption shifted urban children toward the rural centroid, we additionally restricted the analysis to the urban subgroup. Results are provided in Supplementary Data 13.

### Comparison Ethiopia (urban) and Ethiopia (rural) and Italy
To assess differences in gut microbiota composition between study origins (Italy, Ethiopia rural, and Ethiopia urban), we performed non-metric multidimensional scaling (NMDS) and permutational multivariate analysis of variance (PERMANOVA) using the "adonis2" function from the vegan R package. Bray–Curtis dissimilarities were calculated separately from species-level MetaPhlAn profiles and from pathway-level HUMAnN profiles. For each analysis, the proportion of explained variance ($R^2$) and statistical significance (*p*) were determined using 999 permutations.

## AMR analysis

RGI output tables were filtered to retain only high-confidence hits with at least 100 supporting reads and ≥80% breadth of coverage. For each sample, the total number of AMR genes detected was calculated. Group comparisons of AMR gene counts were performed across metadata categories (e.g., age, breastfeeding, delivery mode, teff consumption, IgA-coating bins (samples were grouped into six bins according to their IgA coating frequency; each bin represented an equal fraction of the 0–100% range), household variables, and *H. pylori* status) using Wilcoxon rank-sum tests for binary factors or Kruskal–Wallis tests for variables with more than two groups. *P*-values were adjusted for multiple testing using the BH procedure and are reported in Supplementary Data 12.

For each sample, the number of detected resistance genes was also summarized at the high-level drug-class annotation. Drug-class annotations were harmonized by standardizing the order of multi-class entries so that equivalent categories reported in different orders were treated as the same. Drug-class counts were summed separately for each population group (Ethiopia urban, Ethiopia rural, and Italy). Within each origin, the ten most prevalent drug classes were retained as individual categories, while all remaining classes were grouped as "others."

Alluvial diagrams were created with the easyalluvial R package[81]. Rare categories of AMR genes, resistance mechanisms, and drug classes were combined into an "other" group.

## Ethics and inclusion statement

This study was conducted in accordance with the Global Code of Conduct for Research in Resource-Poor Settings. All ethical regulations relevant to human research participants were followed. The project was performed in collaboration with Ethiopian clinical partner Adugna Negussie, who contributed to study coordination, participant recruitment, and contextual interpretation of the findings and is included as a co-author. The research addressed a locally relevant health topic and was discussed with Ethiopian collaborators prior to study implementation. Roles, responsibilities, and authorship were agreed in advance. The study was approved by the National Ethical Review Board of Ethiopia (ref. no. 3.10/16/2018). Informed consent was obtained from mothers or the legal guardians of each child. All procedures followed international standards for ethical conduct, biosafety, and data protection. The study design and reporting avoided participant stigmatization, and data were analyzed in anonymized form. No exceptional regulatory exemptions were required.

## Reporting summary

Further information on research design is available in the Nature Portfolio Reporting Summary linked to this article.

## Data availability

The sequencing data generated in this study, including metagenomic and processed 16S rRNA reads, are publicly available in the NCBI BioProject database under accession number PRJNA1345963. For metagenomic sequencing, the raw paired-end reads have been deposited without further processing. For 16S rRNA gene sequencing, the deposited data consist of the quality-filtered, merged, and chimera-free reads as provided by the sequencing facility. Source data for all figures are provided in Supplementary Data 14. Sequencing data from external resources are available under accession numbers PRJNA504891 (rural Ethiopian children) and PRJNA716780 (Italian children).

## Code availability

All custom code used for data processing and figure generation is publicly available at GitHub (https://github.com/lydiakirsche/urban-ethiopian-children-microbiome) and archived on Zenodo[82] (10.5281/zenodo.18231174) to ensure long-term accessibility and reproducibility.

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

## Acknowledgements
We are indebted to the participants of this study. We thank Lena Schultheis for help with DNA extraction. This work was supported by the Swiss National Science Foundation (Project grants 320030-236304 and 310030_192490 to A.M.), the Comprehensive Cancer Center Zürich project grant (to A.M., P.L.) the Medical Faculty of the University of Zürich (to A.M.) and the CIFAR Catalyst Fund Project CF-0533 - CP25-090 (to A.M. and M.J.B.). The sponsors had no role in the study design, data analysis or any other part of the research and manuscript submission.

## Author contributions
L.K. performed all metagenomic analyses, generated all figures, and contributed to manuscript writing and revision. P.L. performed QIIME2-based bioinformatics analyses and supported metagenomic data processing. M.J.B. and M.S. provided scientific guidance and critical review. A.N. coordinated patient recruitment and sample collection and contributed to data interpretation. A.M. conceived and supervised the study, secured funding, and led manuscript drafting and finalization. All authors approved the final manuscript.

## Competing interests
The authors declare no competing interests.
