## [Transparent Peer Review File · Communications Biology]

Gut microbial signatures expose the westernized lifestyle of urban Ethiopian children

Corresponding Author: Professor Anne Müller

Version 0:

Reviewer comments:

Reviewer #1

(Remarks to the Author)

This manuscript presents microbiota profiling of urban Ethiopian children using both 16S rRNA sequencing and shotgun metagenomics, with a focus on lifestyle and dietary factors such as teff consumption, H. pylori status, and household crowding. The study is well-executed, and the manuscript is generally clear. However, the findings are largely confirmatory. Some specific points need clarification:

P5: The authors mention random selection of 207 children from a larger birth cohort. How representative is this subset regarding key variables like delivery mode, breastfeeding, nutrition, housing, and medication use? The same applies to subsets used for H. pylori testing, IgA coating, and metagenomics.

P5: The statement about DNA quality being “sufficient despite logistical challenges” needs support. Was DNA integrity or concentration measured? Also, how were fecal samples stored before extraction, were any preservatives or stabilization kits used? This information is important, as storage can affect microbiota profiles (e.g., [PMID: 37106038]).

Figure 1a: How were the Shannon index clusters defined? A brief explanation is needed. Also consider converting Shannon index to effective number of species, which is more interpretable. Unclear abbreviations like SPO should be spelled out.

P8–10: The cross-cohort comparisons (rural Ethiopian and Italian children) lack detail on how the external datasets were processed. Were DNA extraction, sequencing platforms, and bioinformatics pipelines harmonized? Without this, technical biases could undermine conclusions.

P8–10: Given that shotgun metagenomics was performed, it would be useful to report whether any common gut pathogens were detected. This would add relevance to the health implications of the microbiota differences.

P15: Please include summary statistics for the metagenomic data (e.g., read counts, % host reads removed, average coverage).

Reviewer #2

(Remarks to the Author)

I have read with interest the paper entitled “Gut microbial signatures reflect the westernized lifestyle of urban Ethiopian children” by Kirsche et al. The study describes the gut microbiome of Ethiopian children, and compare the effect of rurality versus urbanization. The study is interesting, yet needs to

There are no lines numbering, which makes it more complex

1. The authors describes many factors associated with gut microbiome – the authors would gain in performing univariate analysis such as PERMANOVA on beta-diversity and alpha-diversity– (with variance (R²) and compare the contribution of metadata on gut microbiome variation (see for instance <https://pmc.ncbi.nlm.nih.gov/articles/PMC8754568/>)
2. The authors performed shotgun metagenomic but did not describe well the species level even strain. The introduction mentions “Here, we fill this gap by reporting 16S rRNA and strain-resolved shotgun”
3. Discuss extent of IgA coating versus other studies – and the hypothesis behind the analysis – Can the authors show a distribution of IgA coated bacteria?
4. Which taxa were more found in Teff consumers? what are their main clinical characteristics? Maybe IgA coating is associated with Teff consumption based on other clinical feature
5. The authors should discuss lack of Treponema – even in rural populations?
6. The heatmaps look very complex to visualise messages and could go to supporting information. The authors should find an approach to convey the main messages in the main text –

7. Figure 2 A does not seem to have a message
8. For function related to Teff, authors could look at CAZY

Minor

16S rRNA sequencing – replace by 16S rRNA gene sequencing

Reviewer #3

(Remarks to the Author)

The study is dedicated to children aged 2-5 living in urban areas of Ethiopia. The main discoveries of the study are that urban Ethiopian children lack the microbial signatures of rural Ethiopians related to a plant-based diet, carry more AMR genes, and are generally more similar to European cohorts than to rural Ethiopians. It was found that the consumption of traditional fermented cereals affects the microbiome composition of urban Ethiopians, including diversity.

Major Comment: My main recommendation concerns the Methods section, particularly the statistical analysis. It should be described clearly, step by step. Currently, it is impossible to understand what statistical procedures were used to obtain particular results. Therefore it is difficult for me as a reviewer to determine if all the statistical procedures used were appropriate.

Detailed comments:

1. Introduction

Well written, providing a solid background in the field.

1.1 - "We find that urban Ethiopian children harbor a microbiota that is more closely related to that of Western European children than to that of rural Ethiopians, with the notable absence of *Prevotella*, *Segatella* and other species associated with a non-Western, traditional lifestyle." - I'm not sure it makes sense to include conclusions in the Introduction section.

2. Results

2.1 - It is convenient to include FDR or p-values in the text, not just in the figures. It's also helpful to mention the name of the statistical test (at least once if all results in the paragraph were obtained using the same test).

2.2 - It is known that the number of reads per sample can affect alpha diversity. There are several options to overcome this when comparing samples (for example, correction or rarefaction). Did you apply any of these? If not, it could affect the conclusions. (The same concern - Figure 2b).

2.3 - Figure 1a - The visualization is beautiful and detailed, but I don't grasp the main idea behind it (other than the negative result that clustering was not associated with metadata). Was it supposed to illustrate significant associations between metadata and taxonomic composition? If so, this should be made clearer; currently, the only valuable information I obtained was a list of analyzed metadata factors and top microbes. I believe bar plots showing median abundances of top microbes would be more useful, providing clearer information about the top members of urban Ethiopian children's microbial communities than a heatmap. Additionally, I didn't understand what information the reader could derive from the metadata values shown above the heatmap.

2.4 - Figure 1 b, c, d - Usually, boxplots (violin plots) illustrating non-significant associations are not shown in the main text. I believe that the article would be much clearer and more readable if the figures in the main text included only significant associations. I see no need to show illustrations for negative results.

2.5 - "High abundance of *Bifidobacterium* was associated with low-diversity; the inverse was true for several species including *Faecalibacterium*, *Ruminococcus* and *Anaerostipes*" - What statistical test was used? Was multiple comparison correction applied across taxa? Were the taxa filtered based on abundance or prevalence before the analysis? Was any transformation or correction applied? I did not find this information in the Methods.

2.6 - Figure 2a - Similar comment to Figure 1a. Although the figure is beautiful, I don't understand what information is supposed to be obtained from it. Is age and delivery mode metadata really necessary here? Is it possible to draw any conclusions based on their colors? I recommend simplifying the figure and including only the information that is relevant to the discoveries and revealed associations.

2.7 - Figure 2 legend: "Statistical significance for two-group comparisons was assessed using Mann–Whitney U tests, and for multiple-group comparisons using Kruskal–Wallis tests followed by Dunn's post-hoc tests with multiple testing correction." - I see only two-group comparisons in this figure.

2.8 - Figure 2 - I don't understand what finding is illustrated in the figure. If the intent is to show which COG categories differed between two groups (consuming and not consuming Teff), please mark them with asterisks because I don't see the difference. Perhaps another type of visualization would make the differences clearer?

2.9 - "67 uSGBs-MAGs predominantly found in teff consumers"" - What exactly is meant by "predominantly"? Was this a result of a statistical test? Please explain the procedure more clearly in the Methods section.

2.10 - How was the enrichment of COGs achieved, and what statistical test was used? I see "The y-axis displays the statistical significance as $-\log_{10}$ of the FDR-adjusted p-value," but I do not see the statistical procedure described in the Methods section.

2.11 - "These included *Segatella copri*, *S. sinensis*, *S. hominis*, *S. brasiliensis*, *S. brunsvicensis*, *S. sanihominis* and *S. sinica*, as well as *Faecalibacterium prausnitzii* and *Prevotella* species, all of which were enriched in rural, but not urban Ethiopians". As I understood, were the results obtained with Maaslin? The procedure is not clearly described in the Methods section. Were these results obtained using 16S or WGS data? It is also interesting whether the correction for Teff consumption could affect these associations. Were the children who consume Teff more similar to rural Ethiopians than those who do not?

2.13 - "Metagenomic analyses reveal large differences in the fecal microbiota composition of rural and urban Ethiopians" section - please provide FDR values near corresponding results.

2.14 - Figure 3 legend, Results - "Coefficient plot of pathway abundance associations, displaying the 15 pathways with the highest and lowest β -coefficients, indicating enrichment in either urban or rural children" - This analysis is not described in the Methods section, and I was unable to understand what was done.

2.15 - Figure 4a - Here, the visualization clearly shows the associations between microbiome and children's origin. However, I recommend using asterisks to indicate the taxa for which significant associations were found.

2.16 - Figure 4e - Are the differences significant?

3. Methods

3.1 - Major: The statistical analysis section should be reorganized. It is currently impossible to understand what was done. Please make a separate paragraph for each type of analysis and describe in detail all filtration, transformation, and other steps related to each type, ensuring results could be easily reproduced by other scientists.

3.2 - I have counted 12 metadata factors analyzed for connections to alpha diversity. I believe that multiple comparison problem should be addressed here. I recommend using multiple comparison correction (for example, FDR) in the analysis of associations between metadata factors and alpha diversity (after Mann–Whitney U test for two-group comparisons).

3.3 - I also did not understand if the multiple comparison correction was done only by taxa or by taxa and factors in the Maaslin analysis (I think the second option is correct). Please provide information about the normalizations and transformations used in Maaslin, as there are different options. It is recommended to use CLR transformation when analyzing microbiome data, given that the data are compositional. If you have chosen another option, please explain how you accounted for the compositionality of microbiome data.

3.4 - I recommend adding PERMANOVA analysis using Aitchison distance followed by FDR correction to assess the associations of metadata with microbiome composition in general.

3.5 - "DNA Extraction and 16S rRNA Gene Sequencing" - Qiime2 software and the DADA2 algorithm were used but were not cited. Please check the citation of all software used.

3.6 - "Metagenome sequencing" - please cite the software and algorithms.

3.7 - "All statistical analyses were performed using..." - The word "all" is confusing since Maaslin and other tests were also conducted. Please specify that you are describing associations between metadata and alpha diversity (if I am correct).

3.8 - Fisher's exact test with Baptista–Pike odds ratio estimation was used for categorical comparisons - please specify in detail for what analyses these tests were used.

Reviewer #4

(Remarks to the Author)
Review of CRMICR-D-25-00125

Summary:

This manuscript investigates an important question regarding the impact of urban Western lifestyle on infant gut microbial signatures in a setting typically considered non-Westernized, by characterizing the taxonomic and functional signatures of gut microbiomes in children from a highly urbanized region in East Africa (Adama, Ethiopia). The authors employ both 16S rRNA gene sequencing and metagenomic sequencing alongside relevant demographic and lifestyle data. Based on intra-cohort cross-sectional and longitudinal analyses, complemented by a comparative analysis of another cohort from rural

Ethiopia, the authors report that gut communities of infants living in urbanized Ethiopia share compositional features with those of stereotypical Westernized cohorts (e.g., urban Italy), differing from previously characterized rural African infant gut microbiomes. This work is highly relevant; the authors characterize an underrepresented demographic in microbiome research (African communities exposed to a typical urban Western lifestyle) and expand our understanding of gut microbiome variation across diverse lifestyles. However, the manuscript's conclusions are presently limited by several analytical, methodological, and interpretive issues that need to be addressed to strengthen the results and increase the clarity of the study. Key concerns relate to not appropriately accounting for host age, a major driver of gut community diversity in early infancy, and the limited clarity/reporting of statistical methods and results, among other points detailed below.

Major points:

1) In general, manuscript figures (both in the main text and supplement) aim to convey a large amount of visual information in a relatively small space. Unfortunately, this results in poor readability. Please consider increasing font sizes in all figures.

Also, Figure 1 would benefit from several modifications:

- The heatmap for 16S rRNA gene abundances shows only genus-level annotations, resulting in repeated names for genera like "Bifidobacterium" and "Blautia". Please include the species names or modify the heatmap.
- Please italicize genus and species names in figures and throughout the text.
- The lower panels (b-f), could benefit joint-representation tests such as correlograms, ANOVA, or PERMANOVA with both the 16S rRNA gene and metagenomic taxonomic profiles, rather than multiple univariate tests, improving figure clarity and organization.

2) In Section 1 of Results, the authors divided samples into 6 Shannon Index bins to evaluate the effect of cesarean delivery on diversity. However, it is not clear from either Results or Methods how the authors chose the number of bins or their cutoffs, especially considering the information provided in Fig. 1a. Please include the explanation on why six bins were used. The authors could consider regression models treating Shannon Index as a continuous variable, which would offer a more robust and interpretable analysis and controlling for age. Additionally, a clear account of which bins were compared in that analysis is necessary, provided that Suppl. Figure 1c does not show a clear monotonic trend of decreasing C-section with increasing Shannon bin.

3) Age is a major and well-established factor influencing gut microbiome development; yet, it appears to be underanalyzed and underexplored. It is surprising that age does not appear to significantly influence diversity or composition, despite the trends in Bifidobacterium and Faecalibacterium abundances suggesting otherwise. Several studies on the early-life patterns of gut microbial community succession have established a fundamental relationship between alpha diversity, Bifidobacterium and Faecalibacterium, and host chronology, which is likely at play in this cohort as indicated by Figure 1a (top). Please clarify how age was handled in the analyses, particularly in MaAsLin3 models. Furthermore, the absence of significant age effects but not delivery mode contrasts some of the previous studies (e.g., see Bokulich et al., 2016 PMID: 27306664) that show that the effect of delivery mode in the gut microbiome diminishes as the child approaches the mark of 2 years of age, with other works supporting that this effect is even less relevant after 2 years of age. Thus, differences in alpha diversities among bins require further exploration.

4) Given the availability of both 16S rRNA gene and metagenome data, it is surprising that the initial diversity and compositional analysis (e.g., Figure 1) only contains 16S rRNA gene data. Considering metagenomes for the analyses in Figure 1 would be especially important in the context of alpha diversity, as the two approaches might differ. While 16S rRNA gene data might uncover taxa with low abundance, metagenomic data might uncover taxa that are missed by universal primers. I recommend that authors consider repeating these initial diversity and compositional analyses using the metagenomic taxonomic profiles for improved resolution and robustness.

5) Please report statistical analyses more clearly. Some associations are described qualitatively without corresponding statistics (e.g., effect sizes or p values). The methods section would benefit from expanded detail on statistical models, adjustments, and software usage (including appropriately citing different packages within pipelines such as QIIME, MaAsLin3, or bioBakery tools).

6) Please consider including a table summarizing cohort demographics. Specifically, readers should be able to compare measures like N, mean, SD, max and min (where applicable) or class proportions (where applicable) of relevant metadata such as child age for all of: (1) the original recruited Ethiopian cohort; (2) the subset chosen for 16S sequencing; (3) the subset chosen for metagenomic sequencing - including overlap with (2); (4) the externally sourced Rural Ethiopian cohort; and (5) the externally-sourced Italian cohort.

7) Comparisons between urban and rural cohorts involve significant size imbalances (e.g., 100+ urban vs ~6 rural samples), unclear age matching and limited reporting on covariate balance. One key section of the manuscript is the contrast between urban and rural metagenomes, the latter sourced from a published cohort. This is followed up by comparison with a third, Italian cohort. Those cohorts, however, comprise only around 6 and 15 samples, in contrast with the >100 urban Ethiopian metagenomes analyzed in the work. This raises concerns about the robustness of inter-cohort comparisons. Given the extreme imbalance between urban and rural sizes, the unclear age matching, and the unclear inclusion of age as a factor in the analyses, the authors should further clarify/discuss their analysis with statistical figures and distribution tests. I recommend that authors more explicitly address these cohort imbalances through adjusted models or sensitivity analyses, and report the distributions and age overlap using summary statistics and summarized visualizations.

8) Several papers relevant to your study have not been cited or referenced. Please consider conducting another round of literature search, and including pediatric cohorts that examine urban versus rural gut microbiomes, or the recent publication by Meghini et al. (2025) on "Expanding the human microbiome atlas of Africa" (DOI: 10.1038/s41586-024-08485-8).

9) Sample collection, DNA extraction, sequencing strategies and sequence processing can significantly affect the observed microbial community composition. Please comment on the extent to which the three cohorts were harmonized in terms of these steps and on how this may have impacted the observed results and analyses.

Minor points:

1) Please increase font sizes in figures for readability.

2) Color gradients in Figure 1a-b, for the Shannon Index, lack sufficient contrast. Please consider using color-blind-friendly palettes to increase visual clarity. This also applies for Figure 4.

3) Please specify database versions used for taxonomic (MetaPhlan) and functional (HUMAnN) profiling. Reporting these increases the chances of getting reproducible results.

4) The AMR gene detection results, while interesting, are not well integrated into the narrative and are absent from the Discussion section. Please consider clarifying their relevance or removing them to maintain focus.

5) Phrasing of "known kSGBs" and "unknown uSGBs" could be considered redundant. Please consider rewriting to "known SGBs (kSGBs)" and "unknown SGBs (uSGBs)".

6) Delivery mode abbreviations such as "SPO" (probably meaning spontaneous vaginal delivery) could be replaced with the more broadly accepted SVD or VD (vaginal delivery) to improve clarity. If "SPO" reflects local clinical practice, a brief explanation would be helpful.

7) Figure 4 illustrates three metagenome cohorts side by side. Some of the same issues as noted for Figure 1 apply here. The information is condensed and difficult to read.

8) Please consider including line numbers in your revised document.

Reviewer #5

(Remarks to the Author)

I co-reviewed this manuscript with one of the reviewers who provided the listed reports. This is part of the Communications Biology initiative to facilitate training in peer review and to provide appropriate recognition for Early Career Researchers who co-review manuscripts.

Version 1:

Reviewer comments:

Reviewer #1

(Remarks to the Author)

The authors have adequately addressed my comments.

Reviewer #2

(Remarks to the Author)

I have read the revisions of the paper "Gut microbial signatures reflect the westernized lifestyle of urban Ethiopian children" by Professor Müller and colleagues.

The authors have replied to comments and I don't have any further comments

Line 173 - I find surprising that age does not significantly contribute to gut microbiome variation, especially in infants, but probably the age range is too small (3-4 years)

Reviewer #3

(Remarks to the Author)

Dear Authors,

Thank you for your work! The Methods section is now much clearer. However, I still have one question related to comment 2.5 (copied below). I could not find in the Methods how the "small set of biologically relevant taxa" was defined. Could you please add this information to the Methods?

Here is the original comment and your response:

2.5 - "High abundance of Bifidobacterium was associated with low-diversity; the inverse was true for several species including Faecalibacterium, Ruminococcus and Anaerostipes" - What statistical test was used? Was multiple comparison correction applied across taxa? Were the taxa filtered based on abundance or prevalence before the analysis? Was any transformation or correction applied? I did not find this information in the Methods.

For the analysis, raw counts were transformed to CPM values (with a pseudocount of 1) and collapsed at the genus level. Associations between taxon abundances and Shannon diversity were tested using linear regression models including age as a covariate, with Shannon bins evaluated by F-test. Only a small set of biologically relevant taxa was examined, so no additional filtering or multiple testing correction was applied. These details have now been added to the Methods.

All my other comments have been fully addressed.

Reviewer #4

(Remarks to the Author)

Review of CRMICR-D-25-00125

Summary:

Kirsche and colleagues present a revised version of the manuscript "Gut microbial signatures expose the westernized lifestyle of urban Ethiopian children". The authors have responded thoughtfully and thoroughly to the reviewers' comments, and the revised version has been substantially improved. We appreciate the considerable effort invested in strengthening the methodological description, data visualization and presentation of results, and increasing the transparency of the inter-cohort comparison metric. Nonetheless, a few minor concerns remain, which are outlined in the comments below.

1) Despite the authors' substantial effort to expand and refine the methodology (~40% of the main text now focuses on methods), the rationale for choosing six Shannon Index bins remains unclear. Given that the authors added continuous models for several comparisons (with delivery mode and several taxa), it is difficult to understand the relevance of the Shannon Index bins. If these bins represent biologically informed groupings, the authors should cite past work that derives and justifies this number. If not, the manuscript would benefit from a sensitivity analysis showing how robust the bin-based findings and visualizations are to perturbations in the bin count.

2) The revised manuscript now includes a supplementary table with descriptive statistics for cohort demographics, allowing a high-level comparison of metadata similarity. However, these summaries are essential for interpreting the results. We recommend moving the table to the main text so that readers can readily access and compare the data.

3) Relatedly, the discussion section does not acknowledge any limitations or potential weaknesses of the study. Yet, the summary statistics in Suppl. Table 1 highlights several important considerations: the number of samples across studies is substantially different; the 6 rural samples do not cover the entire age range of the urban samples; and feeding information was not available for every cohort. We suggest adding a concise section to the discussion that describes and addresses how the factors may influence the findings and their potential impact on the interpretation provided in the manuscript.

4) The expanded section on detection and quantification of AMR genes is a welcome addition. However, it remains underdeveloped and somewhat disconnected from the rest of the manuscript (Ln 274). This analysis would be strengthened by clearly identifying the major sources of differences in AMR gene profiles across cohorts and linking them to the broader gut microbiome findings. This would be particularly useful in the context of developing child health.

5) The listed BioProject identifier PRJNA1275869 does not appear to correspond to any data in GenBank. Providing the raw data (e.g., SRA accession numbers) is important for transparency and reproducibility. Also, it would have allowed us to evaluate the dataset.

6) Please provide a link to all code used to generate the data and figures. For code provided through GitHub, we recommend obtaining a persistent identifier (i.e. a DOI) through Zenodo, as described at <https://docs.github.com/en/repositories/archiving-a-github-repository/referencing-and-citing-content>. In the absence of this, please provide the commit hash for the code version used.

Reviewer #5

(Remarks to the Author)

I co-reviewed this manuscript with one of the reviewers who provided the listed reports. This is part of the Communications

Biology initiative to facilitate training in peer review and to provide appropriate recognition for Early Career Researchers who co-review manuscripts.

Version 2:

Reviewer comments:

Reviewer #3

(Remarks to the Author)

Thank you!

All my comments have been fully addressed.

Reviewer #4

(Remarks to the Author)

I have read the revisions of the paper by Kirche and colleagues. Thank you for your clear, point-by-point response. I was able to access sequence reads. Also, thank you for including the sensitivity analysis.

I have no further comments.

Reviewer #5

(Remarks to the Author)

I co-reviewed this manuscript with one of the reviewers who provided the listed reports. This is part of the Communications Biology initiative to facilitate training in peer review and to provide appropriate recognition for Early Career Researchers who co-review manuscripts.

Point-to-point response

Date: 17.10.2025

Title of Article: Gut microbial signatures expose the westernized lifestyle of urban Ethiopian children

Name of the Corresponding Authors: Anne Müller, Lydia Kirsche

Email Address of the Corresponding Authors: mueller@imcr.uzh.ch, kirsche@imcr.uzh.ch

We thank the reviewers and editor for their helpful feedback. In this revised version of the manuscript, we have put considerable effort into clarifying and expanding the methodology and statistical analyses, including more detailed descriptions of the tests used and the reasoning behind them.

We also revised the relevant sections of the results and discussion to address interpretation issues and to make the overall presentation clearer. Figures were updated to improve readability, and we now provide a comprehensive Supplementary Data files in Excel format.

Overall, a lot of work went into strengthening the manuscript, and we believe the revisions significantly improve its robustness, transparency, and clarity.

Please see our answers in green below, alongside the sections of modified text.

Editor:

In particular, all reviewers mentioned the need of much clearer methodology and statistical analysis descriptions. In addition reviewer 2 and 3 raise several result interpretation issues, reviewer 1 would like to see more evidence of the results (e.g. how to show the DNA quality is good), and reviewer 4 identifies poor result readability problems.

For all graphs depicting a single point value (e.g., mean) with error bars, you must add individual data points or convert the graph to a boxplot or dot-plot to show data distribution.

It's mandatory to provide access to the numerical source data for graphs and charts either through a repository or by providing the data in a Supplementary Data file (in excel format).

All blots/gels must be accompanied by size markers in every figure panel. Uncropped and unedited blot/gel images must be included as Supplementary Figure(s) in the Supplementary Information pdf.

We thank you for the constructive feedback. We have expanded the descriptions of methodology and statistical analyses to ensure greater clarity. Data presentation and figure readability were improved, and where possible we used individual data points alongside mean values. An extensive Supplementary Information package of 15 files now accompanies the manuscript, providing detailed supporting analyses underlying the figures. As the study does not include blots or gels, the corresponding requirements are not applicable.

Reviewer #1 (Remarks to the Author):

This manuscript presents microbiota profiling of urban Ethiopian children using both 16S rRNA sequencing and shotgun metagenomics, with a focus on lifestyle and dietary factors such as teff consumption, H. pylori status, and household crowding. The study is well-executed, and the manuscript is generally clear. However, the findings are largely confirmatory.

Some specific points need clarification:

P5: The authors mention random selection of 207 children from a larger birth cohort. How representative is this subset regarding key variables like delivery mode, breastfeeding, nutrition, housing, and medication use? The same applies to subsets used for *H. pylori* testing, IgA coating, and metagenomics.

We have now provided a comparison between the full cohort and the analysed subsets in suppl. Data 1, at least for a subset of parameters of relevance to the study. The text has been modified as follows:

Results, p.4: ...The cohort of 207 children was quite representative of the entire cohort with respect to the age at the time of sampling and the proportion of children born by cesarian section, but showed a more balanced representation of breast-fed and formula-fed children than the total cohort (suppl. Data 1)....

P5: The statement about DNA quality being “sufficient despite logistical challenges” needs support. Was DNA integrity or concentration measured? Also, how were fecal samples stored before extraction, were any preservatives or stabilization kits used? This information is important, as storage can affect microbiota profiles (e.g., [PMID: 37106038]).

The missing information has now been added to the methods section, as follows:

Methods, p.12: ... Immediately after collection, samples were stored at -80°C without the addition of preservatives until further processing. ...

Methods, p.13: ... DNA concentration was determined using a NanoDrop spectrophotometer and only extracts with an A260/280 ratio of ~1.8 and a concentration above 5 ng/μL were considered for downstream analyses.

Figure 1a: How were the Shannon index clusters defined? A brief explanation is needed. Also consider converting Shannon index to effective number of species, which is more interpretable. Unclear abbreviations like SPO should be spelled out.

We now clarify in the figure legend that Shannon index values were divided into six bins of approximately equal size using quantile-based binning, labelled 1-6 from lowest to highest. The effective number of species (Observed ASVs) is provided in suppl. Figure 1i. Following Reviewer 4’s recommendation, the abbreviation “SPO” has been replaced by “SVD” throughout.

Note that the heatmap has been moved from main Figure 1 to supplemental Figure 1b; the supplemental figure legend has been amended with the following statement: ... Samples are binned according to their Shannon diversity index, into six bins of approximately equal size (quantile-based binning), labelled 1-6 from lowest to highest...

P8-10: The cross-cohort comparisons (rural Ethiopian and Italian children) lack detail on how the external datasets were processed. Were DNA extraction, sequencing platforms, and bioinformatics pipelines harmonized? Without this, technical biases could undermine conclusions.

This is very true. We now address this explicitly in the Methods. A new paragraph has been added:

Methods, p.14:

Integration of external metagenomic datasets

For the Italian and rural Ethiopian cohorts, we used publicly available raw sequencing data (Manara et al., 2023; Tett et al., 2019). DNA extraction and library preparation for these datasets had been performed with the same DNeasy® PowerSoil® Pro Kit (#47016, QIAGEN), and NexteraXT DNA Library Preparation Kit (Illumina), followed by sequencing on an Illumina HiSeq2500 platform with 100 nt paired-end reads at a target depth of ~5 Gb/sample. In contrast, our newly generated Ethiopian cohort was sequenced on an Illumina NovaSeq X Plus platform with 150 nt paired-end reads at a target depth of ~12 Gb/sample. To ensure comparability, all raw reads from the external cohorts and our own dataset were re-processed through an identical bioinformatics pipeline (quality control, host filtering,

taxonomic and functional profiling, assembly, binning, and annotation). This harmonization mitigates potential platform-specific effects and enables valid cross-cohort comparisons.

P8-10: Given that shotgun metagenomics was performed, it would be useful to report whether any common gut pathogens were detected. This would add relevance to the health implications of the microbiota differences.

We have now analysed the metagenomic data for common gut pathogens. The results are presented in suppl. Figure 4g. The text has been modified as follows:

Results, p.9: ...Finally, we mined the metagenomes for the (differential) presence of putative bacterial pathogens; indeed, the prevalence of *E. coli*, *Enterococcus*, *Klebsiella pneumoniae* and *Clostridium perfringens* was generally higher in urban Ethiopians than in their rural counterparts, or in Italian children (suppl. Figure 4g). The combined results suggest that urban Ethiopian children have adopted a “westernized” microbiome that is characterized by loss of traditional African signature microbes, a metabolic switch to the utilization of simple refined and dairy sugars, and the acquisition of a diverse set of AMR genes.

Supplemental Figure legends, suppl. Figure 4: ...**(g)** Prevalence of selected potentially pathogenic species detected by MetaPhlAn, stratified by cohort. Prevalence was calculated as the proportion of samples in which each species had non-zero relative abundance. ..

P15: Please include summary statistics for the metagenomic data (e.g., read counts, % host reads removed, average coverage).

We have now provided a comprehensive overview of the metagenomic dataset in suppl. Data 6, including sequencing read statistics, host read removal percentages, and contig assembly statistics, and refer to it in the appropriate section of text.

Results, p.6: ...To gain more in-depth insights into the microbial species of our cohort, and the metabolic functions that might be associated with specific clinical and lifestyle factors, we selected 105 children (suppl. Data 1) among the original cohort of 207 for metagenomic analysis (suppl. Data 6)....

Reviewer #2 (Remarks to the Author):

I have read with interest the paper entitled “Gut microbial signatures reflect the westernized lifestyle of urban Ethiopian children” by Kirsche et al. The study describes the gut microbiome of Ethiopian children, and compare the effect of rurality versus urbanization. The study is interesting, yet needs to be improved. There are no line numbering, which makes it more complex.

We have now introduced line numbering throughout.

1. The authors describes many factors associated with gut microbiome - the authors would gain in performing univariate analysis such as PERMANOVA on beta-diversity and alpha-diversity- (with variance (R²) and compare the contribution of metadata on gut microbiome variation (see for instance <https://pmc.ncbi.nlm.nih.gov/articles/PMC8754568/>)

We thank the reviewer for this helpful suggestion. In line with the comment, we have performed PERMANOVA analyses on β -diversity, testing Teff consumption and all candidate confounders individually, and have included these results in dedicated supplemental datas. None of the factors, including Teff consumption, explained a significant proportion of the variance in gut microbial community structure.

Regarding α -diversity, we respectfully note that PERMANOVA is not applicable, as it is specifically designed for testing effects on multivariate community distance matrices (β -diversity). Since the α -diversity indices are univariate measures, their associations with metadata are more appropriately

assessed using regression models or non-parametric tests. We therefore chose to evaluate α -diversity separately using Wilcoxon/Kruskal-Wallis tests, which are statistically suited for univariate outcomes.

The new supplemental data showing Permanova results are called out in the text as follows:

Results, p.4: ... To assess the impact of metadata factors on overall gut microbiota structure, we first analysed the 16S rRNA gene sequencing data using Bray-Curtis dissimilarities and PERMANOVA, with additional dispersion testing. None of the factors explained a significant proportion of the variance (suppl. Data 2; note that the confounding factor of “wasting” was removed as it showed collinearity with age, suppl. Figure 1a). ..

Results, p.6: ... To assess whether community composition differed across host and environmental factors, we performed PERMANOVA analyses followed by tests for homogeneity of group dispersion. None of the tested variables explained a significant proportion of beta diversity variation (suppl. Data 7). The factor most strongly associated with compositional shifts was frequency of teff consumption, which showed a trend towards significance (PERMANOVA: $R^2 = 0.013$, $p = 0.111$; suppl. Data 7). *H. pylori* infection status accounted for 3.1% of the variance but was not significant ($p = 0.28$). Other factors, including delivery mode, breastfeeding status at 18 months, stunting, age, household size, and the presence of a latrine or separate bedroom, explained $\leq 2.3\%$ of the variance and were non-significant ($p > 0.1$). Tests for dispersion confirmed that none of the associations were confounded by unequal within-group variance (all $p > 0.05$; suppl. Data 7). A closer examination of the α -diversity in relation to individual parameters confirmed some of the trends revealed by the 16S rRNA gene sequencing (suppl. Data 8)....

2. The authors performed shotgun metagenomic but did not describe well the species level even strain. The introduction mentions “Here, we fill this gap by reporting 16S rRNA and strain-resolved shotgun”
Do we understand correctly that our reviewer requests us to correct “strain-resolved” metagenomic analyses to the (indeed more accurate) “species-resolved” metagenomic analyses?

The text has been modified as follows:

Introduction, p.3: ...Here, we fill this gap by reporting 16S rRNA gene sequencing and species-resolved shotgun metagenomic analyses conducted on >200, and >100 young children aged 2-5, respectively, that were born between 2018 and 2022 in Ethiopia’s second largest city, Adama...

3. Discuss extent of IgA coating versus other studies - and the hypothesis behind the analysis - Can the authors show a distribution of IgA coated bacteria?

We thank the reviewer for this valuable comment. Our hypothesis was that Teff consumption influences the extent of IgA coating of intestinal bacteria, reflecting altered mucosal immune-microbiota interactions. To address the request for distribution, we note that the main Figure 2 already presents the range of IgA coating across all tested children. In addition, we now provide suppl.Fig. 1j with three representative flow cytometry plots illustrating low (6.45%), intermediate (40.3%), and high (85.9%) IgA coating. This visualises the variation in IgA-binding patterns at the single-sample level. For context, our observed coating levels are within the range reported by previous flow cytometry-based studies. These are now cited in the discussion as follows:

Discussion, p.11: ...Our levels of IgA coating ($34 \pm 20\%$ of all fecal bacteria) is in the range of on average 10 to 36% reported by others (Eriksen et al., 2023, Catanzaro et al., 2019, Sterlin et al., 2020, Van Der Waaij et al., 2004).

4. Which taxa were more found in Teff consumers? what are their main clinical characteristics? Maybe IgA coating is associated with Teff consumption based on other clinical feature

We examined microbial abundance differences between Teff consumers and non-consumers using MaAsLin3, adjusting for age. Analyses were performed both on the full metagenomic dataset and on

the subset of IgA-positive samples. In neither case did we detect taxa with significant differential abundance between groups. This is now mentioned in the text as shown below.

We further tested whether IgA coating modifies the association between Teff consumption and microbial abundance by including an interaction term (Teff_Freq_Factor_Binary * IgApos_bacteria) in MaAsLin3. This analysis also yielded no significant results. We therefore conclude that Teff consumption is not associated with distinct taxa or IgA coating patterns in our dataset.

Results, p.7: ...Given that Teff consumption was not significantly associated with the prevalence or number of uSGBs, and that no taxa showed differential abundance between teff consumers and non-consumers, we next examined the functional potential of these uSGBs to determine whether teff consumption might instead be linked to differences at the metabolic or pathway level....

5. The authors should discuss lack of *Treponema* - even in rural populations?

We detected three different *Treponema* species in our dataset. These were most prevalent in rural Ethiopian children, whereas they were absent from Italian children and detected only sporadically in urban Ethiopian children. This pattern is in line with previous reports showing that *Treponema* species enrich the gut microbiota of traditional rural populations but are typically absent in urban individuals (Angelakis et al., 2019). The relative abundance of these species in our cohort is shown in suppl. Figure 4d, alongside the following text:

Results, p.9:... Several *Treponema* species associated with traditional lifestyles (Angelakis et al., 2019) did not make the top 20 genera cutoff, but were present in rural Ethiopians, and not the other two cohorts (suppl. Figure 4d).

6. The heatmaps look very complex to visualise messages and could go to supporting information. The authors should find an approach to convey the main messages in the main text -

We agree and have moved the heatmaps to the Supplementary Figures. The main text now highlights the key findings in a simplified manner.

7. Figure 2 A does not seem to have a message

Thank you for pointing this out. We have simplified Figure 2A to focus on the key message: the diversity and number of SGBs reconstructed in our study. Metadata layers (age, delivery mode, QC values) were removed to avoid distraction and are instead described in the Methods. The revised figure now highlights only the taxonomic distribution of SGBs (2B) and the proportion linked to teff consumption, which is the central message we want to show.

8. For function related to Teff, authors could look at CAZY

Thank you for the suggestion. We agree that CAZyme analysis could be informative; however, we addressed carbohydrate-related functions through our COG analysis, which already demonstrates an increase linked to teff consumption.

Minor

16S rRNA sequencing - replace by 16S rRNA gene sequencing

Is replaced!

Reviewer #3 (Remarks to the Author):

The study is dedicated to children aged 2-5 living in urban areas of Ethiopia. The main discoveries of the study are that urban Ethiopian children lack the microbial signatures of rural Ethiopians related to

a plant-based diet, carry more AMR genes, and are generally more similar to European cohorts than to rural Ethiopians. It was found that the consumption of traditional fermented cereals affects the microbiome composition of urban Ethiopians, including diversity.

Major Comment: My main recommendation concerns the Methods section, particularly the statistical analysis. It should be described clearly, step by step. Currently, it is impossible to understand what statistical procedures were used to obtain particular results. Therefore it is difficult for me as a reviewer to determine if all the statistical procedures used were appropriate.

Detailed comments:

1. Introduction

Well written, providing a solid background in the field.

1.1 - "We find that urban Ethiopian children harbor a microbiota that is more closely related to that of Western European children than to that of rural Ethiopians, with the notable absence of *Prevotella*, *Segatella* and other species associated with a non-Western, traditional lifestyle." - I'm not sure it makes sense to include conclusions in the Introduction section.

This is probably a style issue. We prefer to end the introduction section with an outlook on the main results, rather than a vague statement on what was done. We hope this reviewer agrees with us. As the above-stated sentence is the only sentence in the introduction giving away a key result, we think it is justified to leave it there.

2. Results

2.1 - It is convenient to include FDR or p-values in the text, not just in the figures. It's also helpful to mention the name of the statistical test (at least once if all results in the paragraph were obtained using the same test).

We have now pointed out some key p-values in the results section (especially on the 16 S rRNA sequencing data), and have made sure to mention statistical tests in the figure legends.

Examples include:

Results, p.5: ...a-diversity was higher in 3-year old children than in the 4- and 5-year old's, and, surprisingly, in children affected by stunting ($p=0.019$; with stunting defined by the WHO as height-for-age that is more than two standard deviations below the median; Figure 1b). Infection with the gastric pathobiont *H. pylori*, which in East Africa continues to be highly prevalent and which was serologically confirmed in 40% of examined children in our cohort (positive serology increasing with age; suppl. Figure 1f), was linked to lower a-diversity ($p=0.019$; Figure 1c). An important parameter affecting a-diversity was the number of rooms available per family, with a single (over multiple) room(s) being predictive of high diversity ($p=0.002$; Figure 1c). Interestingly, the regular consumption (at least once per week, and up to twice per day) of a locally grown cereal, teff (*Eragrostis tef*) in the form of pancakes ("Injera") or traditional breads made from fermented dough, resulted in a higher diversity of the fecal microbiota ($p=0.05$; Figure 1c, suppl. Figure 1g)...

2.2 - It is known that the number of reads per sample can affect alpha diversity. There are several options to overcome this when comparing samples (for example, correction or rarefaction). Did you apply any of these? If not, it could affect the conclusions. (The same concern - Figure 2b).

We thank the reviewer for raising this important point. To account for differences in sequencing depth, we rarefied all samples to 24,000 reads prior to calculating alpha and beta diversity metrics in QIIME2. This information has been added to methods section as follows:

Methods, p.13: Alpha and beta diversity metrics were computed in QIIME2 to assess microbial community composition and variability after rarefying to 24000 reads.

2.3 - Figure 1a - The visualization is beautiful and detailed, but I don't grasp the main idea behind it (other than the negative result that clustering was not associated with metadata). Was it supposed to illustrate significant associations between metadata and taxonomic composition? If so, this should be made clearer; currently, the only valuable information I obtained was a list of analyzed metadata factors and top microbes. I believe bar plots showing median abundances of top microbes would be more useful, providing clearer information about the top members of urban Ethiopian children's microbial communities than a heatmap. Additionally, I didn't understand what information the reader could derive from the metadata values shown above the heatmap.

To simplify the main figures and emphasise the key findings, Figure 1a has been moved to suppl. Figure 1b. This retains the detailed heatmap and metadata information for interested readers while keeping the main figure focused on the most relevant visualisations.

2.4 - Figure 1 b, c, d - Usually, boxplots (violin plots) illustrating non-significant associations are not shown in the main text. I believe that the article would be much clearer and more readable if the figures in the main text included only significant associations. I see no need to show illustrations for negative results.

We appreciate the reviewer's feedback. We have chosen to retain Figure 1b-d in the main text, as these comparisons provide important exploratory context. In particular, the absence of differences in breastfeeding-associated Shannon diversity and by delivery mode is informative for interpreting our findings and contributes to the overall picture of microbial community variation in this cohort.

2.5 - "High abundance of Bifidobacterium was associated with low-diversity; the inverse was true for several species including Faecalibacterium, Ruminococcus and Anaerostipes" - What statistical test was used? Was multiple comparison correction applied across taxa? Were the taxa filtered based on abundance or prevalence before the analysis? Was any transformation or correction applied? I did not find this information in the Methods.

For the analysis, raw counts were transformed to CPM values (with a pseudocount of 1) and collapsed at the genus level. Associations between taxon abundances and Shannon diversity were tested using linear regression models including age as a covariate, with Shannon bins evaluated by F-test. Only a small set of biologically relevant taxa was examined, so no additional filtering or multiple testing correction was applied. These details have now been added to the Methods.

Methods, p.16: Amplicon sequence variant (ASV) tables and taxonomic assignments were imported from QIIME 2 artifacts. Raw counts were converted to counts per million (CPM) using edgeR(Chen et al., 2025) after adding a pseudocount of 1 to avoid zeros. Target taxa were defined at genus level. For each target taxon, CPM values were summed across all ASVs annotated to that taxon, yielding one CPM value per sample and taxon. Shannon values were divided into six bins (as describes above, labelled 1-6 from lowest to highest). For each bacterial taxon, we used a linear regression model to test whether its CPM abundance was associated with alpha diversity and age. Shannon diversity was included as the 6 bins, and age as a continuous variable. From each model we extracted the effect of age, including the estimate, standard error, and p-value, as well as the overall effect of Shannon diversity, which was tested with an F-test across all six categories (suppl. Data 5). In addition, results from an analogous set of models using continuous Shannon values (instead of bins) are provided (suppl. Data 5).

2.6 - Figure 2a - Similar comment to Figure 1a. Although the figure is beautiful, I don't understand what information is supposed to be obtained from it. Is age and delivery mode metadata really necessary here? Is it possible to draw any conclusions based on their colors? I recommend simplifying the figure and including only the information that is relevant to the discoveries and revealed associations.

Thank you for the comment. We simplified Figure 2a by removing the age, delivery mode, and QC values (contamination and completeness), which are instead described in the Methods. The revised figure now emphasises the key message: the number of SGBs that could be reconstructed and their assignment in relation to teff consumption.

2.7 - Figure 2 legend: "Statistical significance for two-group comparisons was assessed using Mann-Whitney U tests, and for multiple-group comparisons using Kruskal-Wallis tests followed by Dunn's post-hoc tests with multiple testing correction." - I see only two-group comparisons in this figure.

The figure legends have been rewritten and now provide much more detail on all statistical tests used. In the case of Figure 2, the relevant text is as follows:

Statistical significance was assessed using Wilcoxon rank-sum tests.

2.8 - Figure 2 - I don't understand what finding is illustrated in the figure. If the intent is to show which COG categories differed between two groups (consuming and not consuming Teff), please mark them with asterisks because I don't see the difference. Perhaps another type of visualization would make the differences clearer?

We thank the reviewer for this helpful observation. The intention of the figure was to illustrate the distribution of COG categories identified in the uSGBs. We agree that it does not convey a key difference relevant to the main storyline. To improve the clarity and focus of the main text, we have therefore moved this figure to suppl. Fig 2a.

2.9 - "67 uSGBs-MAGs predominantly found in teff consumers"" - What exactly is meant by "predominantly"? Was this a result of a statistical test? Please explain the procedure more clearly in the Methods section.

We thank the reviewer for this valuable comment. The term "predominantly" was imprecise and has therefore been removed from the manuscript. In this analysis, we considered all uSGBs identified in the dataset.

2.10 - How was the enrichment of COGs achieved, and what statistical test was used? I see "The y-axis displays the statistical significance as $-\log_{10}$ of the FDR-adjusted p-value," but I do not see the statistical procedure described in the Methods section.

Thanks, this is now added to the methods section:

Methods, p.18: ... To identify features differing in relative abundance between Teff consumers and non-consumers, we compared distributions using the Wilcoxon rank-sum test. For each feature present in both groups, a p-value was obtained and adjusted for multiple testing using the Benjamini-Hochberg false discovery rate (FDR). COG categories with $FDR < 0.05$ were considered significant (suppl. Data 11).

For visualization, we computed the \log_2 fold change in median relative abundance between consumers and non-consumers. Volcano plots were generated with \log_2 fold change on the x-axis and $-\log_{10}(FDR\text{-adjusted } p)$ on the y-axis. Significant categories were highlighted based on FDR only (no fold-change cutoff was applied).

2.11 - "These included *Segatella copri*, *S. sinensis*, *S. hominis*, *S. brasiliensis*, *S. brunsvicensis*, *S. sanihominis* and *S. sinica*, as well as *Faecalibacterium prausnitzii* and *Prevotella* species, all of which were enriched in rural, but not urban Ethiopians". As I understood, were the results obtained with Maaslin? The procedure is not clearly described in the Methods section. Were these results obtained using 16S or WGS data? It is also interesting whether the correction for Teff consumption could affect these associations. Were the children who consume Teff more similar to rural Ethiopians than those who do not?

We thank the reviewer for pointing this out. We have now clarified in the Methods that the reported taxonomic associations were obtained from metagenomic data using the MetaPhlAn/HUMAnN pipeline and analysed with MaAsLin3. The relevant procedures are now described in detail in the revised Methods section. In addition, we investigated whether Teff consumption influenced these associations by testing for shifts in overall similarity to rural microbiomes. While Teff consumption showed a consistent but non-significant trend toward greater similarity with rural children, it did not really alter the main associations.

Methods, p.17: Associations between genus-level microbial abundances and host metadata were tested using MaAsLin3 (Nickols et al., 2024), which fits multivariable regression models for each microbial feature while controlling for covariates. Genus-level abundance tables were obtained from collapsed ASV counts and filtered to include only genera with non-zero variance. For each analysis, we tested the effect of the variable of interest (*H. pylori* status, Stunting, Teff consumption, Rooms in home) on microbial abundance while controlling for age as a continuous covariate (mean-centered, Age_c). Mean centering was applied to improve model stability and interpretability of the coefficients. All models were run with total-sum scaling (TSS) normalization and log transformation of microbial abundances, as implemented in MaAsLin3. Statistical significance was defined at FDR < 0.05 using MaAsLin3's Benjamini-Hochberg-adjusted term-level q-values. No significant associations were detected in any of the models. To visualise effect size distributions, we therefore plotted the genera with the 10 highest and 10 lowest β -coefficients from each analysis, together with 95% confidence intervals and corresponding FDR-adjusted values.

2.13 - "Metagenomic analyses reveal large differences in the fecal microbiota composition of rural and urban Ethiopians" section - please provide FDR values near corresponding results.

This problem has now been dealt with. Instead of FDR values, we are including measures of robustness. This is explained in the Methods section and supplemental figure legend as follows:

Methods, p.19: ...To assess robustness, we performed a resampling stability analysis. In each of 100 iterations, we included all 6 rural samples and a random subset of 25 urban samples, re-fitting the same MaAsLin3 model. For each contrast, we recorded the proportion of iterations with $q < 0.05$ (stability of significance) and the proportion of iterations in which the effect sign was the same (directional consistency). A species was considered "Robust" if it reached significance in $\geq 25\%$ of iterations and exhibited a consistent effect direction in $\geq 75\%$ of iterations. For visualisation, significant species from the full model ($q < 0.05$) were displayed, and robustness was indicated by bar opacity.

Figure legend, suppl. Figure 3: ...Diverging plot of species-level abundance differences between urban ($n=105$) and rural Ethiopian ($n=6$) children. Bars show MaAsLin3 β -coefficients for species significant in the full model ($q < 0.05$), adjusted for centered age. Positive values indicate higher abundance in urban samples and negative values indicate higher abundance in rural samples. Opacity encodes Robustness status from the resampling analysis (25 urban vs 6 rural per iteration; 100 iterations): Robust (criterion met: significant in $\geq 25\%$ of iterations with $\geq 75\%$ consistent effect sign) at full opacity, Not robust at reduced opacity. Species are ordered by effect size.

2.14 - Figure 3 legend, Results - "Coefficient plot of pathway abundance associations, displaying the 15 pathways with the highest and lowest β -coefficients, indicating enrichment in either urban or rural

children" - This analysis is not described in the Methods section, and I was unable to understand what was done.

We agree that the pathway analysis was insufficiently described in the Methods. We have now added the following description:

Methods, p.19: For pathway analysis, we applied Maaslin3 using Age as a continuous covariate, as described above. The 15 significant pathways with the highest and lowest β -coefficients for abundance in relation to Origin were plotted.

2.15 - Figure 4a - Here, the visualization clearly shows the associations between microbiome and children's origin. However, I recommend using asterisks to indicate the taxa for which significant associations were found.

We thank the reviewer for this suggestion. To improve clarity, Figure 4a has been moved to the Supplement.

2.16 - Figure 4e - Are the differences significant?

No, they are not significant.

3. Methods

3.1 - Major: The statistical analysis section should be reorganized. It is currently impossible to understand what was done. Please make a separate paragraph for each type of analysis and describe in detail all filtration, transformation, and other steps related to each type, ensuring results could be easily reproduced by other scientists.

We have now fully rewritten the Statistical Analysis section. It is organised into separate paragraphs for each type of analysis, with detailed descriptions of all filtering, transformation, and processing steps to facilitate reproducibility.

3.2 - I have counted 12 metadata factors analyzed for connections to alpha diversity. I believe that multiple comparison problem should be addressed here. I recommend using multiple comparison correction (for example, FDR) in the analysis of associations between metadata factors and alpha diversity (after Mann-Whitney U test for two-group comparisons).

We have now assessed potential collinearity between metadata factors using Cramer's V statistic and excluded redundant variables (Cramer's $V > 0.7$). For beta-diversity, we tested associations with metadata using PERMANOVA (adonis2, 999 permutations) on Bray-Curtis distances and verified homogeneity of group dispersions (betadisper/permutest). For alpha-diversity we applied Wilcoxon rank-sum tests for two-level factors and Kruskal-Wallis tests with post-hoc Dunn's tests for multi-level factors. To account for multiple testing, Benjamini-Hochberg FDR correction was applied across all factors and post-hoc comparisons. All analyses were performed for both 16S rRNA gene and shotgun metagenomic data.

3.3 - I also did not understand if the multiple comparison correction was done only by taxa or by taxa and factors in the Maaslin analysis (I think the second option is correct). Please provide information about the normalizations and transformations used in Maaslin, as there are different options. It is recommended to use CLR transformation when analyzing microbiome data, given that the data are compositional. If you have chosen another option, please explain how you accounted for the compositionality of microbiome data.

We acknowledge that centered log-ratio (CLR) transformation is often recommended for compositional microbiome data. In our analysis, we used the default Maaslin3 pipeline with total sum scaling (TSS) normalization followed by log transformation. This approach reduces skewness and is

widely used in Maaslin-based microbiome studies, ensuring comparability with previously published work.

3.4 - I recommend adding PERMANOVA analysis using Aitchison distance followed by FDR correction to assess the associations of metadata with microbiome composition in general.

For beta-diversity analysis, we used Bray-Curtis dissimilarity combined with the Adonis (PERMANOVA) test. This choice was motivated by two reasons:

(1) Bray-Curtis is a non-Euclidean distance metric that is well-suited for relative abundance data and is widely applied in microbiome studies, allowing comparability across published work.

(2) The Adonis implementation in the vegan R package provides a robust, permutation-based test for assessing the association of metadata with overall community composition.

We acknowledge that Aitchison distance, which relies on CLR-transformed data, directly accounts for the compositional nature of microbiome data and can provide complementary insights. However, given that our main association testing in MaAsLin3 was also based on TSS-normalized and log-transformed abundances, we selected Bray-Curtis to maintain methodological consistency between the univariate and multivariate analyses.

3.5 - "DNA Extraction and 16S rRNA Gene Sequencing" - Qiime2 software and the DADA2 algorithm were used but were not cited. Please check the citation of all software used.

Is cited now!

3.6 - "Metagenome sequencing" - please cite the software and algorithms.

Is cited now!

3.7 - "All statistical analyses were performed using..." - The word "all" is confusing since Maaslin and other tests were also conducted. Please specify that you are describing associations between metadata and alpha diversity (if I am correct).

Is corrected now!

3.8 - Fisher's exact test with Baptista-Pike odds ratio estimation was used for categorical comparisons - please specify in detail for what analyses these tests were used.

We corrected this and specified the correct tests.

Methods, p18:... To assess whether Teff consumption frequency differed by age, we compared age groups using Fisher's exact test. Because only one child in the 5-year-old group reported Teff consumption, we additionally performed a 2x2 Fisher's exact test contrasting 5-year-olds with all younger children (2-4 years). Effect sizes are reported as odds ratios with 95% confidence intervals in suppl. Data 10.

Reviewer #4 (Remarks to the Author):

Review of CRMICR-D-25-00125

Summary:

This manuscript investigates an important question regarding the impact of urban Western lifestyle on infant gut microbial signatures in a setting typically considered non-Westernized, by characterizing the taxonomic and functional signatures of gut microbiomes in children from a highly urbanized region in

East Africa (Adama, Ethiopia). The authors employ both 16S rRNA gene sequencing and metagenomic sequencing alongside relevant demographic and lifestyle data. Based on intra-cohort cross-sectional and longitudinal analyses, complemented by a comparative analysis of another cohort from rural Ethiopia, the authors report that gut communities of infants living in urbanized Ethiopia share compositional features with those of stereotypical Westernized cohorts (e.g., urban Italy), differing from previously characterized rural African infant gut microbiomes. This work is highly relevant; the authors characterize an underrepresented demographic in microbiome research (African communities exposed to a typical urban Western lifestyle) and expand our understanding of gut microbiome variation across diverse lifestyles. However, the manuscript's conclusions are presently limited by several analytical, methodological, and interpretive issues that need to be addressed to strengthen the results and increase the clarity of the study. Key concerns relate to not appropriately accounting for host age, a major driver of gut community diversity in early infancy, and the limited clarity/reporting of statistical methods and results, among other points detailed below.

Major points:

1) In general, manuscript figures (both in the main text and supplement) aim to convey a large amount of visual information in a relatively small space. Unfortunately, this results in poor readability. Please consider increasing font sizes in all figures.

Font size has now been increased throughout!

Also, Figure 1 would benefit from several modifications:

- The heatmap for 16S rRNA gene abundances shows only genus-level annotations, resulting in repeated names for genera like "Bifidobacterium" and "Blautia". Please include the species names or modify the heatmap.

- Please italicize genus and species names in figures and throughout the text.

- The lower panels (b-f), could benefit joint-representation tests such as correlograms, ANOVA, or PERMANOVA with both the 16S rRNA gene and metagenomic taxonomic profiles, rather than multiple univariate tests, improving figure clarity and organization.

We thank the reviewer for these helpful suggestions.

The heatmap has been moved to the Suppl. Fig 1b. Instead of displaying the top 50 ASVs, we now show the top 40 genera, aggregated at genus level. This avoids redundancy from multiple ASVs/species assigned to the same genus. Genus names have been italicized in all figures and throughout the text. Genus and species names are now italicized throughout.

Regarding the statistical analyses, we chose to retain the multiple univariate tests for the main figure, as this presentation allows clearer interpretation of individual associations. However, we now provide extensive supplemental data tables reporting PERMANOVA results for beta diversity and FDR-corrected alpha diversity values for both 16S rRNA gene and metagenomic profiles.

We decided to retain 16S rRNA gene data as the main focus in Figure 1, since this dataset contains nearly twice the number of samples compared to the metagenomic dataset, thereby providing more bigger statistical power.

2) In Section 1 of Results, the authors divided samples into 6 Shannon Index bins to evaluate the effect of cesarean delivery on diversity. However, it is not clear from either Results or Methods how the authors chose the number of bins or their cutoffs, especially considering the information provided in Fig. 1a. Please include the explanation on why six bins were used. The authors could consider regression models treating Shannon Index as a continuous variable, which would offer a more robust and interpretable analysis and controlling for age. Additionally, a clear account of which bins were compared in that analysis is necessary, provided that Suppl. Figure 1c does not show a clear monotonic trend of decreasing C-section with increasing Shannon bin.

We thank the reviewer for this valuable comment. We have now added a detailed description in the Methods section on how the Shannon bins were defined. In our main analysis, we chose to present Shannon diversity in six bins, as we found that this representation makes the interpretation of a complex continuous variable more intuitive, particularly when communicating the results across groups. The number of six bins was chosen arbitrarily, but the general approach of stratifying Shannon indices into categories is well established in the microbiome literature, where tertiles are frequently used (e.g., (Casas et al., 2019; Menni et al., 2017; Zhang et al., 2025))

Methods, p.16: Associations between microbial diversity and delivery mode were assessed in children aged 2-5 years with available delivery information. Alpha diversity (Shannon index) was taken from the Qiime2 output. Shannon indices were divided into six equal-count bins using the `cut_number` function from ggplot2 package(Wickham, 2016) in R, and the proportion of C-section births was calculated within each bin and age group. Logistic regression models with delivery mode (C-section vs spontaneous vaginal delivery) as the outcome and age as a covariate were then fitted. In the binned model, Shannon diversity was included as a categorical predictor (six bins, lowest bin as reference), with odds ratios and 95% confidence intervals estimated for each bin and a likelihood ratio χ^2 test used to assess the global association. In a second model, Shannon diversity was analysed as a continuous predictor, yielding an age-adjusted odds ratio per unit increase in Shannon index with Wald test p-value. All results are reported in suppl. Data 4.

To address the reviewer's concern, we also performed regression models treating Shannon diversity as a continuous predictor, adjusted for age, and present all of these results in suppl. Data 4. While the continuous analysis yields comparable conclusions, we decided to retain the binned representation as the main analysis for clarity and accessibility.

3) Age is a major and well-established factor influencing gut microbiome development; yet, it appears to be underanalyzed and underexplored. It is surprising that age does not appear to significantly influence diversity or composition, despite the trends in Bifidobacterium and Faecalibacterium abundances suggesting otherwise. Several studies on the early-life patterns of gut microbial community succession have established a fundamental relationship between alpha diversity, Bifidobacterium and Faecalibacterium, and host chronology, which is likely at play in this cohort as indicated by Figure 1a (top). Please clarify how age was handled in the analyses, particularly in MaAsLin3 models. Furthermore, the absence of significant age effects but not delivery mode contrasts some of the previous studies (e.g., see Bokulich et al., 2016 PMID: 27306664) that show that the effect of delivery mode in the gut microbiome diminishes as the child approaches the mark of 2 years of age, with other works supporting that this effect is even less relevant after 2 years of age. Thus, differences in alpha diversities among bins require further exploration.

We thank the reviewer for this important point. Age was now included as a covariate in all regression models of Shannon diversity and delivery mode. In the main models using binned Shannon diversity, none of the examined taxa showed significant associations with age, indicating that age did not confound the bin-based analyses. When modelling Shannon as a continuous predictor, some taxa showed modest age-related trends, with Faecalibacterium and Ruminococcus both increasing significantly with age ($p=0.036$ and $p=0.048$, respectively), whereas Bifidobacterium displayed a non-significant negative trend (added in suppl. Data 5). We therefore conclude that within our restricted age range (2-5 years), age was not a dominant driver of the observed associations, although the observed trends are consistent with established patterns of microbial maturation in childhood.

Methods, p.16: Amplicon sequence variant (ASV) tables and taxonomic assignments were imported from QIIME 2 artifacts. Raw counts were converted to counts per million (CPM) using edgeR(Chen et al., 2025) after adding a pseudocount of 1 to avoid zeros. Target taxa were defined at genus level. For

each target taxon, CPM values were summed across all ASVs annotated to that taxon, yielding one CPM value per sample and taxon. Shannon values were divided into six bins (as describes above, labelled 1-6 from lowest to highest. For each bacterial taxon, we used a linear regression model to test whether its CPM abundance was associated with alpha diversity and age. Shannon diversity was included as the 6 bins, and age as a continuous variable. From each model we extracted the effect of age, including the estimate, standard error, and p-value, as well as the overall effect of Shannon diversity, which was tested with an F-test across all six categories (suppl. Data 5). In addition, results from an analogous set of models using continuous Shannon values (instead of bins) are provided (suppl. Data 5).

4) Given the availability of both 16S rRNA gene and metagenome data, it is surprising that the initial diversity and compositional analysis (e.g., Figure 1) only contains 16S rRNA gene data. Considering metagenomes for the analyses in Figure 1 would be especially important in the context of alpha diversity, as the two approaches might differ. While 16S rRNA gene data might uncover taxa with low abundance, metagenomic data might uncover taxa that are missed by universal primers. I recommend that authors consider repeating these initial diversity and compositional analyses using the metagenomic taxonomic profiles for improved resolution and robustness.

In addition to the 16S rRNA gene-based analyses presented in Figure 1 and suppl. Data 3, we have performed the same diversity and compositional analyses using the metagenomic taxonomic profiles. These results are now included in suppl. Data 8.

5) Please report statistical analyses more clearly. Some associations are described qualitatively without corresponding statistics (e.g., effect sizes or p values). The methods section would benefit from expanded detail on statistical models, adjustments, and software usage (including appropriately citing different packages within pipelines such as QIIME, MaAslin3, or bioBakery tools).

We have now expanded the Methods to give clearer details on the statistical models, adjustments, and software used. We also added the corresponding effect sizes and p-values where associations were previously only described qualitatively. In addition, we now cite the different tools and packages applied (QIIME2, MaAsLin3, bioBakery, R packages).

6) Please consider including a table summarizing cohort demographics. Specifically, readers should be able to compare measures like N, mean, SD, max and min (where applicable) or class proportions (where applicable) of relevant metadata such as child age for all of: (1) the original recruited Ethiopian cohort; (2) the subset chosen for 16S sequencing; (3) the subset chosen for metagenomic sequencing - including overlap with (2); (4) the externally sourced Rural Ethiopian cohort; and (5) the externally-sourced Italian cohort.

A comprehensive demographic summary table has been added as suppl. Data 1, covering the original Ethiopian cohort, the 16S and metagenomic subsets (including overlap), and the externally sourced rural Ethiopian and Italian cohorts.

The text has been modified as follows:

Results, p.4: ... The cohort of 207 children was quite representative of the entire cohort with respect to the age at the time of sampling and the proportion of children born by cesarian section, but showed a more balanced representation of breast-fed and formula-fed children than the total cohort (suppl. Data 1)....

7) Comparisons between urban and rural cohorts involve significant size imbalances (e.g., 100+ urban vs ~6 rural samples), unclear age matching and limited reporting on covariate balance. One key section of the manuscript is the contrast between urban and rural metagenomes, the latter sourced from a published cohort. This is followed up by comparison with a third, Italian cohort. Those cohorts, however, comprise only around 6 and 15 samples, in contrast with the >100 urban Ethiopian

metagenomes analyzed in the work. This raises concerns about the robustness of inter-cohort comparisons. Given the extreme imbalance between urban and rural sizes, the unclear age matching, and the unclear inclusion of age as a factor in the analyses, the authors should further clarify/discuss their analysis with statistical figures and distribution tests. I recommend that authors more explicitly address these cohort imbalances through adjusted models or sensitivity analyses, and report the distributions and age overlap using summary statistics and summarized visualizations.

We appreciate the reviewer's concern regarding sample size imbalances and age distribution across cohorts. We agree that these are important limitations, particularly in the rural Ethiopian cohort, which was restricted by the availability of publicly accessible data. To address this, all Maaslin3 analyses now include Age as a continuous covariate, thereby accounting for potential age-related effects. Summary statistics for sample size and age distribution of each cohort are reported in suppl. Data 1.

While the small sample size of the rural cohort limits statistical power, these comparisons were intended as exploratory and hypothesis-generating. To further assess robustness, we repeated the analyses using downsampled subsets of the urban cohort. Figure 3a has been adapted accordingly, highlighting the species that remain robustly significant across iterations and those that should be considered exploratory. We have also revised the manuscript text to explicitly acknowledge these limitations and clarify the exploratory nature of the inter-cohort contrasts.

8) Several papers relevant to your study have not been cited or referenced. Please consider conducting another round of literature search, and including pediatric cohorts that examine urban versus rural gut microbiomes, or the recent publication by Meghini et al. (2025) on "Expanding the human microbiome atlas of Africa" (DOI: 10.1038/s41586-024-08485-8).

This very interesting paper has now been cited several times in the appropriate sections of the manuscript. Several other studies have been identified and included as well.

9) Sample collection, DNA extraction, sequencing strategies and sequence processing can significantly affect the observed microbial community composition. Please comment on the extent to which the three cohorts were harmonized in terms of these steps and on how this may have impacted the observed results and analyses.

We now address this explicitly in the Methods. A new paragraph has been added:

Methods, p.14: For the Italian and rural Ethiopian cohorts, we used publicly available raw sequencing data. DNA extraction and library preparation for these datasets had been performed with the same DNeasy® PowerSoil® Pro Kit (#47016, QIAGEN), and NexteraXT DNA Library Preparation Kit (Illumina), followed by sequencing on an Illumina HiSeq2500 platform with 100 nt paired-end reads at a target depth of ~5 Gb/sample. In contrast, our newly generated Ethiopian cohort was sequenced on an Illumina NovaSeq X Plus platform with 150 nt paired-end reads. To ensure comparability, all raw reads from the external cohorts and our own dataset were re-processed through an identical bioinformatics pipeline (quality control, host filtering, taxonomic and functional profiling, assembly, binning, and annotation). This harmonization mitigates potential platform-specific effects and enables valid cross-cohort comparisons.

Minor points:

1) Please increase font sizes in figures for readability.

Sizes were increased!

2) Color gradients in Figure 1a-b, for the Shannon Index, lack sufficient contrast. Please consider using color-blind-friendly palettes to increase visual clarity. This also applies for Figure 4.

Thank you, colour blind friendly colours were used!

3) Please specify database versions used for taxonomic (MetaPhlan) and functional (HUMAnN) profiling. Reporting these increases the chances of getting reproducible results.

Databases are added!

4) The AMR gene detection results, while interesting, are not well integrated into the narrative and are absent from the Discussion section. Please consider clarifying their relevance or removing them to maintain focus.

We have now included them in the Discussion section as follows:

Discussion, p.12: The mining of metagenomic data for AMR genes allowed us to quantitatively annotate the gene counts on the one hand and on the other hand assign drug classes corresponding to these genes. Whereas the gene counts did not differ significantly between rural and urban populations, the diversity of drug classes clearly differentiated urban and rural cohorts. This finding is in line with published reports of other geographical areas (Balachandra et al., 2021; Du et al., 2025).

5) Phrasing of “known kSGBs” and “unknown uSGBs” could be considered redundant. Please consider rewriting to “known SGBs (kSGBs)” and “unknown SGBs (uSGBs)”.

Corrected!

6) Delivery mode abbreviations such as “SPO” (probably meaning spontaneous vaginal delivery) could be replaced with the more broadly accepted SVD or VD (vaginal delivery) to improve clarity. If “SPO” reflects local clinical practice, a brief explanation would be helpful.

SPO has been replaced by SVD throughout, thank you for this helpful recommendation.

7) Figure 4 illustrates three metagenome cohorts side by side. Some of the same issues as noted for Figure 1 apply here. The information is condensed and difficult to read.

We thank the reviewer for this comment. To improve clarity, the heatmap of Figure 4 has been moved to the Supplement.

8) Please consider including line numbers in your revised document.

Line numbering is now used throughout the document.

References

- Angelakis, E., Bachar, D., Yasir, M., Musso, D., Djossou, F., Gaborit, B., Brah, S., Diallo, A., Ndombe, G. M., Mediannikov, O., Robert, C., Azhar, E. I., Bibi, F., Nsana, N. S., Parra, H. J., Akiana, J., Sokhna, C., Davoust, B., Dutour, A., & Raoult, D. (2019). Treponema species enrich the gut microbiota of traditional rural populations but are absent from urban individuals. *New Microbes and New Infections*, 27, 14–21. <https://doi.org/10.1016/J.NMNI.2018.10.009>
- Balachandra, S. S., Sawant, P. S., Huilgol, P. G., Vithya, T., Kumar, G., & Prasad, R. (2021). Antimicrobial resistance (AMR) at the community level. *Journal of Family Medicine and Primary Care*, 10(3), 1404–1411. https://doi.org/10.4103/JFMPC.JFMPC_888_20
- Casas, L., Karvonen, A. M., Kirjavainen, P. V., Täubel, M., Hyytiäinen, H., Jayaprakash, B., Lehmann, I., Standl, M., Pekkanen, J., & Heinrich, J. (2019). Early life home microbiome and hyperactivity/inattention in school-age children. *Scientific Reports 2019 9:1*, 9(1), 1–9. <https://doi.org/10.1038/s41598-019-53527-1>
- Chen, Y., Chen, L., Lun, A. T. L., Baldoni, P. L., & Smyth, G. K. (2025). edgeR v4: powerful differential analysis of sequencing data with expanded functionality and improved support for small counts and larger datasets. *Nucleic Acids Research*, 53(2). <https://doi.org/10.1093/NAR/GKAF018>
- Du, S., Tang, H., Wang, Z., Chen, H., Fang, X., Sun, J., Niu, Z., Liu, Y., Hu, Y., Su, W., Zhang, Z., Prapamontol, T., Nakayama, S. F., Huang, J., Norback, D., Wu, Q., Tan, Y., & Zhao, Z. (2025). Children’s home environments as reservoirs of antimicrobial resistance: Divergent urban-rural risks from antibiotic resistance genes and pathogens. *Journal of Hazardous Materials*, 495, 139053. <https://doi.org/10.1016/J.JHAZMAT.2025.139053>
- Manara, S., Selma-Royo, M., Huang, K. D., Asnicar, F., Armanini, F., Blanco-Miguez, A., Cumbo, F., Golzato, D., Manghi, P., Pinto, F., Valles-Colomer, M., Amoroso, L., Corrias, M. V., Ponzoni, M., Raffaetà, R., Cabrera-Rubio, R., Olcina, M., Pasolli, E., Collado, M. C., & Segata, N. (2023). Maternal and food microbial sources shape the infant microbiome of a rural Ethiopian population. *Current Biology*, 33(10), 1939-1950.e4. <https://doi.org/10.1016/J.CUB.2023.04.011>
- Menni, C., Jackson, M. A., Pallister, T., Steves, C. J., Spector, T. D., & Valdes, A. M. (2017). Gut microbiome diversity and high-fibre intake are related to lower long-term weight gain. *International Journal of Obesity (2005)*, 41(7), 1099. <https://doi.org/10.1038/IJO.2017.66>
- Nickols, W. A., Kuntz, T., Shen, J., Maharjan, S., Mallick, H., Franzosa, E. A., Thompson, K. N., Nearing, J. T., & Huttenhower, C. (2024). MaAsLin 3: Refining and extending generalized multivariable linear models for meta-omic association discovery. *BioRxiv*, 2024.12.13.628459. <https://doi.org/10.1101/2024.12.13.628459>
- Tett, A., Huang, K. D., Asnicar, F., Fehlner-Peach, H., Pasolli, E., Karcher, N., Armanini, F., Manghi, P., Bonham, K., Zolfo, M., De Filippis, F., Magnabosco, C., Bonneau, R., Lusingu, J., Amuasi, J., Reinhard, K., Rattei, T., Boulund, F., Engstrand, L., ... Segata, N. (2019). The Prevotella copri Complex Comprises Four Distinct Clades Underrepresented in Westernized Populations. *Cell Host & Microbe*, 26(5), 666-679.e7. <https://doi.org/10.1016/J.CHOM.2019.08.018>
- Wickham, H. (2016). *ggplot2: Elegant Graphics for Data Analysis*. <https://ggplot2.tidyverse.org>.
- Zhang, H. R., Tian, W., Qi, G. X., Zhou, B. Sen, & Sun, Y. J. (2025). Diet-microbiome synergy: unraveling the combined impact on frailty through interactions and mediation. *Nutrition Journal*, 24(1), 135. <https://doi.org/10.1186/S12937-025-01201-W>

Point-to-point response

Date: 01.12.2025

Title of Article: Gut microbial signatures expose the westernized lifestyle of urban Ethiopian children

Name of the Corresponding Authors: Anne Müller, Lydia Kirsche

Email Address of the Corresponding Authors: mueller@imcr.uzh.ch, kirsche@imcr.uzh.ch

We appreciate the reviewers' comments and have made the necessary revisions. Please see our answers in green below, alongside the sections of modified text.

Reviewer #1 (Remarks to the Author):

The authors have adequately addressed my comments.

We thank the reviewer for the positive feedback and are pleased that the comments have been fully addressed.

Reviewer #2 (Remarks to the Author):

I have read the revisions of the paper "Gut microbial signatures reflect the westernized lifestyle of urban Ethiopian children" by Professor Müller and colleagues.

The authors have replied to comments and I don't have any further comments

Line 173 - I find surprising that age does not significantly contribute to gut microbiome variation, especially in infants, but probably the age range is too small (3-4 years)

We thank the reviewer for the positive assessment! We appreciate the note regarding age-related microbiome variation; indeed, the narrow age range might limit our ability to detect age-associated effects.

Reviewer #3 (Remarks to the Author):

Dear Authors,

Thank you for your work! The Methods section is now much clearer. However, I still have one question related to comment 2.5 (copied below). I could not find in the Methods how the "small set of biologically relevant taxa" was defined. Could you please add this information to the Methods?

Here is the original comment and your response:

2.5 - "High abundance of Bifidobacterium was associated with low-diversity; the inverse was true for several species including Faecalibacterium, Ruminococcus and Anaerostipes" - What statistical test was used? Was multiple comparison correction applied across taxa? Were the taxa filtered based on

abundance or prevalence before the analysis? Was any transformation or correction applied? I did not find this information in the Methods.

For the analysis, raw counts were transformed to CPM values (with a pseudocount of 1) and collapsed at the genus level. Associations between taxon abundances and Shannon diversity were tested using linear regression models including age as a covariate, with Shannon bins evaluated by F-test. Only a small set of biologically relevant taxa was examined, so no additional filtering or multiple testing correction was applied. These details have now been added to the Methods.

All my other comments have been fully addressed.

Thank you for pointing this out. We have now clarified in the Methods how the “small set of biologically relevant taxa” was defined. Specifically, the selected genera were chosen because they ranked within the top 20 genera by abundance and variability, showed significant associations with diversity in exploratory analyses, and are well established in SCFA-related gut ecology. We have added this information to the Methods, and we are pleased to hear that all other comments have been fully addressed to your satisfaction.

Methods, p.17:...A targeted set of genera (*Bifidobacterium*, *Faecalibacterium*, *Ruminococcus*, *Agathobacter*, *Anaerostipes*, [*Ruminococcus*] *gouvrenauii* group, and *Fusicatenibacter*) was selected based on significant associations with diversity in exploratory analyses, high abundance and variability (top 20 genera), and established roles in SCFA-related gut ecology.

Reviewer #4 (Remarks to the Author):

Review of CRMICR-D-25-00125

Summary:

Kirsche and colleagues present a revised version of the manuscript “Gut microbial signatures expose the westernized lifestyle of urban Ethiopian children”. The authors have responded thoughtfully and thoroughly to the reviewers’ comments, and the revised version has been substantially improved. We appreciate the considerable effort invested in strengthening the methodological description, data visualization and presentation of results, and increasing the transparency of the inter-cohort comparison metric. Nonetheless, a few minor concerns remain, which are outlined in the comments below.

1) Despite the authors’ substantial effort to expand and refine the methodology (~40% of the main text now focuses on methods), the rationale for choosing six Shannon Index bins remains unclear. Given that the authors added continuous models for several comparisons (with delivery mode and several taxa), it is difficult to understand the relevance of the Shannon Index bins. If these bins represent biologically informed groupings, the authors should cite past work that derives and justifies this number. If not, the manuscript would benefit from a sensitivity analysis showing how robust the bin-based findings and visualizations are to perturbations in the bin count.

As noted in the first revision, the purpose of binning was not inferential but communicative. The binned representation provides an intuitive visualisation of how selected species relate to the microbial diversity across the cohort, as it simplifies a continuous metric into easily interpretable categories.

To address the reviewer’s suggestion more comprehensively, we have now conducted a sensitivity analysis in which the bin-based analyses was repeated using 4, 5, 6, 7, and 8 bins. The result of this

has been added to Supplementary Data 5. Across bin counts, the qualitative patterns and statistical conclusions remained unchanged, demonstrating that our findings are robust to the choice of bin number.

In parallel, and as already included in the previous revision, we also report regression models treating Shannon diversity as a continuous variable (Supplementary Data 5). These models yield conclusions that are fully consistent with the binned results.

Now, the manuscript now includes both continuous and categorical analyses, as well as a detailed sensitivity assessment of the binning strategy. We therefore retain the six-bin visualisation in the main text, as we believe it provides the clearest and most interpretable representation of the data, while all supporting analyses are transparently documented in the supplementary material.

Methods, p.17:... In addition, results from an analogous set of models using continuous Shannon values (instead of bins), as well as a sensitivity analysis repeating the bin-based model using 4-8 bins, are provided (suppl. Data 5).

2) The revised manuscript now includes a supplementary table with descriptive statistics for cohort demographics, allowing a high-level comparison of metadata similarity. However, these summaries are essential for interpreting the results. We recommend moving the table to the main text so that readers can readily access and compare the data.

We thank the reviewer for this suggestion. We agree that descriptive statistics are essential, and for that reason we provide them in Supplementary Table 1. We prefer to keep this table in the Supplement, as the two additional cohorts are introduced and discussed in different sections, and integrating a combined table into the main text would disrupt the flow and structure of the manuscript.

3) Relatedly, the discussion section does not acknowledge any limitations or potential weaknesses of the study. Yet, the summary statistics in Suppl. Table 1 highlights several important considerations: the number of samples across studies is substantially different; the 6 rural samples do not cover the entire age range of the urban samples; and feeding information was not available for every cohort. We suggest adding a concise section to the discussion that describes and addresses how the factors may influence the findings and their potential impact on the interpretation provided in the manuscript.

We have now added a paragraph to the Discussion that outlines the main limitations of the study.

p.12: While the analyses yield consistent and biologically plausible patterns, several practical considerations should be noted. Sample sizes differed across the urban, rural, and Italian cohorts, and the rural metagenomic subset was comparatively small, which may limit the detection of more subtle differences. In addition, the rural samples did not fully match the age range of the urban cohort, and feeding information was not available across all cohorts. These factors mainly affect cross-cohort comparisons, whereas the intra-cohort associations were examined within a well-powered and well-characterised data set. We therefore interpret between-cohort contrasts cautiously and regard them as exploratory patterns that can guide future, more targeted investigations.

4) The expanded section on detection and quantification of AMR genes is a welcome addition. However, it remains underdeveloped and somewhat disconnected from the rest of the manuscript (Ln 274). This analysis would be strengthened by clearly identifying the major sources of differences

in AMR gene profiles across cohorts and linking them to the broader gut microbiome findings. This would be particularly useful in the context of developing child health.

We thank the reviewer for this constructive recommendation. We agree that identifying the major drivers of AMR gene differences would be valuable; however, the available metadata do not allow us to reliably assess potential explanatory factors across the three cohorts. While we were able to explore metadata in detail within the urban cohort (see suppl. Figure 2g, suppl. Data 12), comparable information (e.g., rooms in home or dietary habits) was not available for the rural and Italian datasets. To avoid overinterpretation, we therefore refrain from speculating on causes of AMR variation beyond the descriptive comparisons presented in the Results. We have, however, added a brief statement to the Results to more clearly link the AMR profiles to the cohorts, and we address the broader implications and especially limitation in the Discussion.

p9:... These AMR gene and drug class profiles represent an additional dimension along which the three cohorts differ.

p12:... Differences in AMR profiles are often shaped by differences in antibiotic exposure and healthcare, and it is plausible that such factors contribute to the patterns observed here. (Fredriksen *et al.*, 2023) However, comparable metadata on antibiotic use, treatment history, or healthcare utilisation were not available for the cohorts, preventing a direct assessment of these potential drivers. Within these constraints, the AMR gene repertoire nevertheless provides an additional dimension along which the analysed populations diverge.

5) The listed BioProject identifier PRJNA1275869 does not appear to correspond to any data in GenBank. Providing the raw data (e.g., SRA accession numbers) is important for transparency and reproducibility. Also, it would have allowed us to evaluate the dataset.

Thank you for pointing this out. The BioProject identifier had to be updated due to an internal issue at the NIH. The correct BioProject accession is PRJNA1345963, which now includes all raw sequencing data (including SRA files). A reviewer link is available here: <https://dataview.ncbi.nlm.nih.gov/object/PRJNA1345963?reviewer=1vvsb9h23ji6tfdisd2ln3gsi7>. The full dataset will be released upon publication. We have updated the manuscript accordingly

Data availability, p.22: The sequencing data generated in this study, including metagenomic and processed 16S rRNA reads, are publicly available in the NCBI BioProject database under accession number PRJNA1345963. For metagenomic sequencing, the raw paired-end reads have been deposited without further processing. For 16S rRNA gene sequencing, the deposited data consist of the quality-filtered, merged, and chimera-free reads as provided by the sequencing facility. Source data for all figures are provided with this paper.

6) Please provide a link to all code used to generate the data and figures. For code provided through GitHub, we recommend obtaining a persistent identifier (i.e. a DOI) through Zenodo, as described at <https://docs.github.com/en/repositories/archiving-a-github-repository/referencing-and-citing-content>. In the absence of this, please provide the commit hash for the code version used.

Thank you for your note. We would like to clarify that no custom code, algorithms, or newly developed computational methods were used in this study. All analyses relied exclusively on publicly available R packages and standard functions (e.g., tidyverse, ggplot2). The used R scripts consisted only of normal routine data handling and figure-generation steps and are not “previously unreported

custom computer code” or “central to the main claims”, as defined in the Communications Biology editorial policies.

Because of that, the conditions for mandatory code sharing (custom code that is central to the conclusions) do not apply. For this reason, we do not plan to include a code repository or DOI-linked code. All relevant analytical procedures are already fully described in the Methods.

We are of course happy to clarify specific parameters or function calls upon request.

Reviewer #5 (Remarks to the Author):

I co-reviewed this manuscript with one of the reviewers who provided the listed reports. This is part of the Communications Biology initiative to facilitate training in peer review and to provide appropriate recognition for Early Career Researchers who co-review manuscripts.

References

Fredriksen, S. *et al.* (2023) "Resistome expansion in disease-associated human gut microbiomes," *Microbiome*, 11(1), pp. 166-. Available at: <https://doi.org/10.1186/S40168-023-01610-1>.